# SAFETY-PRIORITIZING CURRICULA FOR CONSTRAINED REINFORCEMENT LEARNING

**Cevahir Koprulu**[1]* **Thiago D. Simão**[2] **Nils Jansen**[3] **Ufuk Topcu**[1]
[1]The University of Texas at Austin    [2]Eindhoven University of Technology
[3]Ruhr-University Bochum

## ABSTRACT

Curriculum learning aims to accelerate reinforcement learning (RL) by generating curricula, i.e., sequences of tasks of increasing difficulty. Although existing curriculum generation approaches provide benefits in sample efficiency, they overlook safety-critical settings where an RL agent must adhere to safety constraints. Thus, these approaches may generate tasks that cause RL agents to violate safety constraints during training and behave suboptimally after. We develop a safe curriculum generation approach (SCG) that aligns the objectives of constrained RL and curriculum learning: improving safety during training and boosting sample efficiency. SCG generates sequences of tasks where the RL agent can be safe and performant by initially generating tasks with minimum safety violations over high-reward ones. We empirically show that compared to the state-of-the-art curriculum learning approaches and their naively modified safe versions, SCG achieves optimal performance and the lowest amount of constraint violations during training.

## 1 INTRODUCTION

Curriculum learning for reinforcement learning (RL) aims to generate task sequences that boost the performance and speed of convergence of RL agents (Narvekar et al., 2020). A common strategy in curriculum generation is to start with easy tasks and adjust the difficulty toward the target tasks as the RL agent improves. Automating curriculum generation increases sample efficiency in wide-ranging environments (Baranes & Oudeyer, 2010; Jiang et al., 2021b) with minimum human effort.

The typical exploration by trial-and-error in RL may cause unsafe behaviors during training, making such techniques unsuitable for safety-critical scenarios (Kendall et al., 2019). In addition to improving sample efficiency, curriculum learning can potentially mitigate this issue by prioritizing tasks with no or low potential for harm so that an RL agent can learn how to accomplish a task without behaving unsafely (Turchetta et al., 2020). For example, a curriculum can start by proposing to an agent learning how to drive a traffic scene without cars and pedestrians, minimizing the risk of accidents.

Constrained RL addresses safety-critical scenarios where, given a safety threshold, a constraint on a cost function characterizes safe behavior (Altman, 1999). A constrained RL agent aims to maximize its reward while satisfying such constraint (Achiam et al., 2017). A standard metric for safety violations during training is the *constraint violation regret*, i.e., accumulated excess cost over the safety threshold (Efroni et al., 2020). Curriculum learning approaches overlook constrained RL and fail to consider the cost constraint. They cannot distinguish unsafe behaviors, and propose tasks that, while yielding high rewards, also incur high costs, leading to high constraint violation regret.

A standard curriculum learning method aims to help an RL agent achieve higher rewards faster. Thus, a naive combination of an off-the-shelf curriculum learning approach and a constrained RL algorithm fails to minimize constraint violation regret due to their *misaligned objectives*. In comparison, a constrained RL algorithm searches for policies that primarily satisfy the cost constraint while maximizing reward as much as possible. Given such a combination, the curriculum generator can propose a task that allows the agent to collect high rewards and simultaneously costs higher than the safety threshold, which violates the constraint. Therefore, we argue that a safe automated curriculum generation method should actively prioritize tasks aligned with both objectives.

---

*Correspondence to: Cevahir Koprulu (cevahir.koprulu@utexas.edu). Code

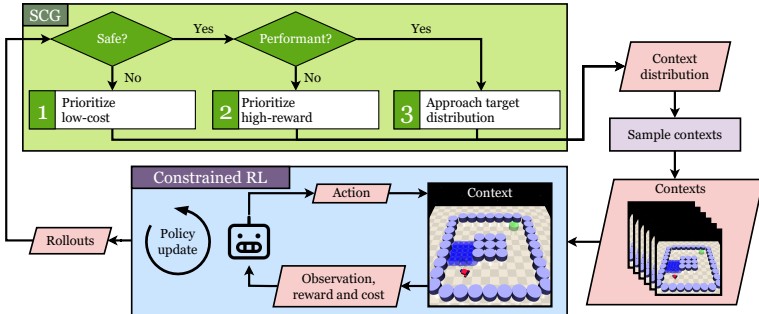

Figure 1: SCG initially prioritizes low-cost contexts to minimize safety violations, then high-reward contexts to boost performance, and finally approaches target distribution by treating them equally.

We propose a *safe* curriculum generation method (SCG) that improves performance, accelerates learning, and minimizes safety violations during training (Fig. 1). Inspired by CURROT (Klink et al., 2022), given a distribution over target tasks, SCG generates a sequence of task distributions that allows the current policy to collect higher rewards than a performance threshold and lower costs than a safety threshold. Initially, SCG prioritizes safety over performance by proposing tasks where the agent satisfies a cost constraint. Once the agent behaves safely in all possible tasks under the current distribution, SCG shifts its focus to satisfying a performance constraint. After the agent becomes performant in all contexts in the current support, SCG generates task distributions that approach the target distribution by equally treating safety and performance until the end of the training.

**Contribution.** Our contribution is three-fold: 1) We describe how existing curriculum learning approaches fail to learn an optimal behavior in a constrained environment safely in Section 4.2, 2) propose **S**afe **C**urriculum **G**eneration (SCG), an automated curriculum learning approach developed for constrained RL to boost learning speed and minimize constraint violation regret in Section 5, and lastly 3) our empirical results provide evidence that, compared to the state-of-the-art curriculum generation approaches and their naively modified versions that account for safety, SCG achieves optimal behavior with the lowest constraint violation regret in constrained RL environments Section 6.

## 2  RELATED WORK

**Curriculum learning for RL.** Automated curriculum generation in RL aims to accelerate convergence to optimal policies by changing the environment configuration according to agent performance. A typical curriculum is a sequence of distributions over such configurations. Florensa et al. (2017) propose generating distributions over initial states where, early on during the training, the agent starts nearby the goal. Other works focus on goal states by optimizing for value disagreement (Zhang et al., 2020), intrinsic motivation (Baranes & Oudeyer, 2010; Portelas et al., 2020), feasibility and coverage of goal states (Racaniere et al., 2020), and intermediate task difficulty (Florensa et al., 2018; Tzannetos et al., 2023). Dennis et al. (2020) propose unsupervised level design as an alternative curriculum learning paradigm and an approach that adversarially generates environment configurations while avoiding infeasible ones. Jiang et al. (2021b;a) study generating distributions over levels, namely, environment instances, that allow the agent to have high learning potential. Due to its recent success, we study *self-paced RL*, a method adopted from supervised learning to order training samples in increasing complexity (Kumar et al., 2010; Jiang et al., 2015). Eimer et al. (2021) generate sequences of tasks with a high capacity for value improvement. Ren et al. (2018) minimizes coverage penalty by generating sequences of environment interactions. Klink et al. (2020a;b; 2021; 2022); Koprulu & Topcu (2023); Koprulu et al. (2023); Huang et al. (2022) formulate the curriculum generation problem as interpolations between task distributions. In contrast, Chen et al. (2021) proposes a gradient-based exploration method via variational inference to expand the task distribution to the entire task space.

**Curriculum learning for safety during training.** Most curriculum learning approaches overlook the safety aspect of RL. Nevertheless, there are methods akin to curriculum learning to increase safety during training. Song & Schneider (2022) propose a genetic algorithm to generate curricula that improve robustness but does not consider an explicit notion of cost nor study environments that highlight the misalignment between performance and safety. Wang et al. (2022) develop a curriculum-

guided RL approach for real-time bidding systems that relaxes cost constraints to incentivize safe policies during training. Eysenbach et al. (2018) learn a reset policy that interferes with the training to prevent the agent from entering dangerous states. Similarly, Turchetta et al. (2020) learn a curriculum policy that chooses an intervention that takes the agent to a safe state if it enters a trigger state. Eysenbach et al. (2018)'s approach trains the reset policy and the RL agent together, whereas Turchetta et al. (2020) consecutively train multiple students to learn an optimal curriculum policy. In comparison, existing curriculum learning methods do not interfere with the interactions between the student and the environment but only assume that a teacher can set the environment configuration for which the agent learns an optimal behavior (Florensa et al., 2017; 2018; Portelas et al., 2020; Jiang et al., 2021a;b; Klink et al., 2020a;b; 2021; 2022). Similarly, SCG does not assume control over environment dynamics, even when the student violates the constraint. Tomilin et al. (2025) propose a new benchmark for constrained RL and shows the benefits of using a handcrafted curriculum.

**Constrained RL.** Constrained RL studies safety-critical settings where errors during exploration may cause constraint violations (Kendall et al., 2019; Roy et al., 2022; Kamran et al., 2022). Therefore, a constrained RL approach aims to achieve safe behavior during and after training (Müller et al., 2024; Simão et al., 2021). Constrained RL approaches that guarantee zero safety violation during training propose using Gaussian processes as transition models (Sui et al., 2015; Berkenkamp et al., 2017; Turchetta et al., 2019; Wachi & Sui, 2020), Lyapunov functions for ensuring global constraints (Chow et al., 2018), or formal methods (Junges et al., 2016; Alshiekh et al., 2018; Jansen et al., 2020). To address high dimensional state and action spaces, Achiam et al. (2017); Tessler et al. (2019); Yang et al. (2020); Hogewind et al. (2022) develop safe policy search algorithms with soft guarantees of not violating the constraints, whereas Achiam & Amodei (2019) integrates a Lagrangian approach into popular RL algorithms. Similar to our setting of interest, Lin et al. (2024) addresses constrained contextual MDPs, where the context determines the dynamics, yet the reward and cost functions are fixed. Furthermore, Lin et al. (2024) focuses on the safety aspect of offline RL and does not study curriculum learning. Another line of work introduces a framework where dynamics, reward, and cost functions remain the same, yet safety thresholds vary (Yao et al., 2023; Günster et al., 2024).

## 3 BACKGROUND AND PROBLEM STATEMENT

We formulate the environments of interest as contextual constrained Markov decision processes to model a constrained multi-task setting given a distribution over target contexts.

### 3.1 CONTEXTUAL CONSTRAINED MDPs

**Definition 3.1.** *We define a* contextual constrained Markov decision process *(CCMDP)* $\mathcal{M} = \langle \mathcal{S}, \mathcal{A}, \mathcal{X}, \mathsf{M}, D, \gamma \rangle$ *with a state space* $\mathcal{S}$*, an action space* $\mathcal{A}$*, a context space* $\mathcal{X} \subseteq \mathbb{R}^n$ *for* $n \in \mathbb{Z}^+$*, a mapping from context space to constrained Markov decision process parameters* $\mathsf{M}$*, a safety threshold* $D \in \mathbb{R}_{\geq 0}$*, and a discount factor* $\gamma \in [0, 1]$*.*

A CCMDP $\mathcal{M}$ represents a family of constrained MDPs parameterized by their contexts $\mathbf{x} \in \mathcal{X}$. A context $\mathbf{x}$ provides a constrained MDP $\mathsf{M}(\mathbf{x}) = \langle \mathcal{S}, \mathcal{A}, p_{\mathbf{x}}, r_{\mathbf{x}}, c_{\mathbf{x}}, p_{0,\mathbf{x}}, \gamma \rangle$, where $\mathcal{S}$, $\mathcal{A}$, and $\gamma$ are the same as in $\mathcal{M}$, but its probabilistic transition function $p_{\mathbf{x}} \colon \mathcal{S} \times \mathcal{A} \to \Delta(\mathcal{S})$, reward function $r_{\mathbf{x}} \colon \mathcal{S} \times \mathcal{A} \to \mathbb{R}$, cost function $c_{\mathbf{x}} \colon \mathcal{S} \times \mathcal{A} \to \mathbb{R}_{\geq 0}$, and initial state distribution $p_{0,\mathbf{x}} \in \Delta(\mathcal{S})$ depend on its context $\mathbf{x}$. A policy $\pi \colon \mathcal{S} \times \mathcal{A} \times \mathcal{X} \to \Delta(\mathcal{A})$ defines the behavior of an agent in a CCMDP $\mathcal{M}$ as a probability simplex over the action space $\mathcal{A}$ given $\mathbf{s} \in \mathcal{S}$ and $\mathbf{x} \in \mathcal{X}$. Note that the agent observes the context $\mathbf{x}$. Following policy $\pi$, an agent collects a trajectory $\boldsymbol{\tau}_{\mathbf{x}} = \{(\mathbf{s}_t, \mathbf{a}_t, r_t, c_t)\}_{t=0}^{T}$ of length $T$ with an initial state $\mathbf{s}_0 \sim p_{0,\mathbf{x}}$, states $\mathbf{s}_{t+1} \sim p_{\mathbf{x}}(\cdot|\mathbf{s}_t, \mathbf{a}_t)$, actions $\mathbf{a}_t \sim \pi(\cdot|\mathbf{s}_t, \mathbf{x})$, rewards $r_t = r_{\mathbf{x}}(\mathbf{s}_t, \mathbf{a}_t)$, and costs $c_t = c_{\mathbf{x}}(\mathbf{s}_t, \mathbf{a}_t)$ at time steps $t \in [T]$, resulting in a discounted cumulative reward and cost $G_r(\boldsymbol{\tau}_{\mathbf{x}}) = \sum_{t=0}^{T} \gamma^t r_t$ and $G_c(\boldsymbol{\tau}_{\mathbf{x}}) = \sum_{t=0}^{T} \gamma^t c_t$, respectively.

Given a CCMDP $\mathcal{M}$ and a target context distribution $\varphi$, i.e., a probability simplex $\Delta(\mathcal{X})$, *contextual constrained RL* aims to maximize expected return subject to a cost constraint:

$$\pi^* \doteq \arg\max_{\pi} \mathbb{E}_{\varphi}[V_r^{\pi}(\mathbf{x})], \quad \text{s.t.} \quad \mathbb{E}_{\varphi}\left[V_c^{\pi}(\mathbf{x})\right] \leq D, \tag{1}$$

where $V_r^{\pi} = \mathbb{E}_{\pi, p_{\mathbf{x}}, p_{0,\mathbf{x}}}[G_r(\boldsymbol{\tau}_{\mathbf{x}})]$ and $V_c^{\pi} = \mathbb{E}_{\pi, p_{\mathbf{x}}, p_{0,\mathbf{x}}}[G_r(\boldsymbol{\tau}_{\mathbf{x}})]$ are the expected discounted cumulative reward and cost, respectively, induced by policy $\pi$ in context $\mathbf{x}$ drawn from $\varphi$.

Fig. 2 shows *safety-maze*, a constrained version of maze (Klink et al., 2022) with simple dynamics on 2D state and action spaces, namely, the agent's coordinates and its displacement, respectively. Starting from the bottom left corner (green), the agent must reach a goal while avoiding the hazards (red), where it collects cost. The context specifies the goal position and tolerance, i.e., the Euclidean distance to the goal for success. The agent can move freely over the white areas but cannot access the walled sections in black. Note that the goal can be positioned over the walls. Safety threshold $D$ determines how much the agent needs to avoid the hazards. An episode terminates when the agent reaches the goal. The target context distribution is uniform over the top white horizontal area.

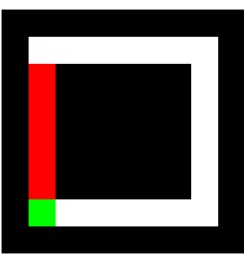

Figure 2: *Safety-maze.*

## 3.2 CONTEXTUAL CONSTRAINED RL

Contextual-constrained RL (CCRL) is an online multi-task-constrained RL framework that does not assume access to the transition, reward, and cost functions. As the optimal policy maximizes the expected discounted return while satisfying a cost constraint, a CCRL algorithm should focus on sample efficiency as well as safety. One metric to measure safety is *constraint violation regret*, the difference between the safety threshold and the value of a learned policy (Efroni et al., 2020). Given that a CCRL algorithm $\Lambda$ runs for $L$-many episodes during training, we define the training regret as

$$\text{Reg}^{tr}(L, \{\varrho_l\}_{l=1}^L, D) \doteq \sum_{l \in [L]} \left[ \mathbb{E}_{\varrho_l}[V_c^{\pi_l}(\mathbf{x})] - D \right]_+ , \qquad (2)$$

where $[y]_+ = \max\{y, 0\}$, $\pi_l$ refers to the policy at the $l^{\text{th}}$ episode, and $\varrho_l$ is the context distribution from which $\mathbf{x}$ is drawn at episode $l$. The regret is non-zero only when the expected discounted cumulative cost exceeds the safety threshold $D$. Thus, training regret $\text{Reg}^{tr}$ only considers the *safety violations* of an algorithm $\Lambda$ with respect to context distributions $\{\varrho_l\}_{l=1}^L$. Once $\Lambda$ converges to an optimal policy, its training regret also converges. Our problem is learning an optimal policy and achieving it with the minimum constrained violation regret. Section 4.2 describes how naive designs of $\Lambda$ fail in this problem due to misaligned objectives of curriculum learning and constrained RL.

**Problem statement.** Given a CCMDP $\mathcal{M}$ to describe the parameterization of a set of constrained tasks and a target context distribution $\varphi$ to specify their probability of occurrence, generate a sequence of context distributions $\{\varrho_l\}_{l=1}^L$ that allow an RL agent to *sample-efficiently* learn an optimal policy (1) with minimal constraint violation regret (2) by taking the misalignment phenomenon into account.

Traditionally, a curriculum learning approach generates a sequence of context distributions $\{\varrho_l\}_{l=1}^L$, while a non-curriculum approach draws contexts directly from the target context distribution. Thus, curriculum learning approaches can choose a context distribution $\varrho_l$ prioritizing contexts where the current policy has low expected cost $V_c^{\pi_l}(\mathbf{x})$ to minimize constraint violation regret.

## 4 CURRICULUM LEARNING AND CONSTRAINED RL

We present a state-of-the-art curriculum learning method and discuss its limitations in CCRL.

### 4.1 CURRICULA VIA OPTIMAL TRANSPORT

Curricula via Optimal Transport (CURROT, Klink et al., 2022), given a target context distribution $\varphi$, creates a sequence of context distributions $\{\varrho_k\}_{k=0}^K$ to obtain an optimal policy for a contextual MDP $\tilde{\mathcal{M}} = \langle \mathcal{S}, \mathcal{A}, \mathcal{X}, \tilde{\mathsf{M}}, \gamma \rangle$ (Hallak et al., 2015). Compared to a contextual constrained MDP, a contextual MDP $\tilde{\mathcal{M}}$ does not have a *cost* function, as $\tilde{\mathsf{M}}(\mathbf{x}) = \langle \mathcal{S}, \mathcal{A}, p_\mathbf{x}, r_\mathbf{x}, p_{0,\mathbf{x}}, \gamma \rangle$. An optimal policy $\pi^*$ in a contextual MDP $\tilde{\mathcal{M}}$ only maximizes the expected return, i.e., $\pi^* \doteq \arg\max_\pi \mathbb{E}_\varphi[V_r^\pi(\mathbf{x})]$.

At curriculum iteration $k \in [K]$, CURROT draws contexts $\{\mathbf{x}_i\}_{i=0}^M$ from context distribution $\varrho_{k-1}$, and collects trajectories $\mathcal{D}_k = \{\boldsymbol{\tau}_{\mathbf{x}_i}\}_{i=1}^M$, where $\boldsymbol{\tau}_{\mathbf{x}_i} = \{(\mathbf{s}_{i,t}, \mathbf{a}_{i,t}, r_{i,t}, \mathbf{s}_{i,t+1})\}_{t=0}^{|\boldsymbol{\tau}_{\mathbf{x}_i}|}$. Then, an RL algorithm updates policy $\pi_{k-1}$ via $\mathcal{D}_k$. CURROT generates the next context distribution via

$$\arg\min_\varrho \ \mathcal{W}_2(\varrho, \varphi) \quad \text{s.t.} \quad \varrho(\mathbf{x}) > 0 \Rightarrow V_r^{\pi_k}(\mathbf{x}) \geq \zeta, \forall \mathbf{x} \in \mathcal{X}, \text{ and } \mathcal{W}_2(\varrho, \varrho_+) \leq \epsilon, \qquad (3)$$

where $\mathcal{W}_2(\cdot, \cdot)$ is the Wasserstein distance and $\varrho_+$ is a particle-based distribution based on contexts with return $G_r(\tau_{\mathbf{x}})$ higher than performance threshold $\zeta$, which a buffer of successful contexts $\mathcal{B}_+$ keeps. A failure buffer $\mathcal{B}_-$ in parallel maintains the remaining contexts. CURROT estimates $V_r^{\pi_k}$ based on $\mathcal{B}_-$ and $\mathcal{B}_+$. The constraint on $V_r^{\pi_k}$ ensures support over contexts with sufficiently high rewards. The Wasserstein distance constraint avoids diverging from the current successful contexts. We investigate CURROT due to its favorable properties: 1) It poses curriculum generation as a constrained optimization problem, enabling a natural extension to CCRL (1). 2) Interpolating context distributions based on Wasserstein distance allows non-parametric distributions. 3) The performance constraint enforces robustness. We refer the reader to Klink et al. (2022) for more details.

## 4.2 FAILURE OF CURRICULA TO ENSURE SAFETY

The state-of-the-art curriculum learning methods, e.g., CURROT focus on the standard multi-task RL problem, i.e., maximizing $\mathbb{E}_\varphi[V_r^\pi(\mathbf{x})]$. They fail to address CCRL due to prioritizing contexts $\mathbf{x} \sim \varrho_k$ where policy $\pi_k$ achieves high $V_r^\pi(\mathbf{x})$ but violate the constraint on $V_c^\pi(\mathbf{x})$. Imagine safety-maze with safety threshold $D = 0$ and the initial *easy* contexts are around the bottom left corner. As the agent improves, CURROT will generate a context distribution closer to the target distribution. However, as CURROT minimizes Wasserstein distance, it will move its contexts over the hazards. Although they set goals closer to the initial position, hence high $V_r^\pi$, they can cause high $V_c^\pi$ and constraint violations. The agent will choose to pass through the hazards or stay out. The former scenario leads to unsafe behavior with high costs, whereas the latter yields failed conservative behavior.

Fig. 3 demonstrates curricula generated by CURROT and SCG in *safety-maze*. The marked points visualize the contexts, determining the goal location and tolerance, drawn from context distributions $\varrho_k$ at different curriculum iterations/epochs. The color of a point refers to the goal tolerance. CURROT moves

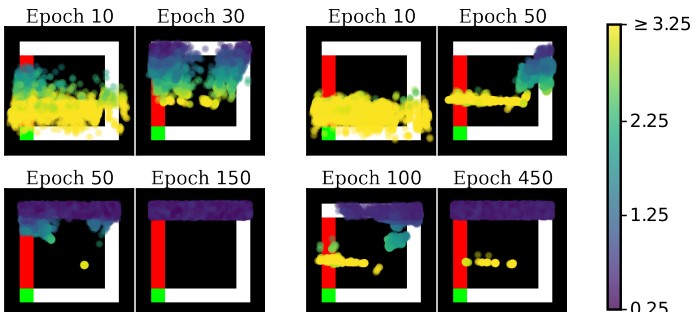

Figure 3: CURROT (left) and SCG's (right) curricula.

contexts from the bottom row toward the target context distribution. As CURROT ignores the cost, it places the goals mostly in the red region early on, causing suboptimal behaviors. In comparison, SCG enables goals centered on the right column, which does not yield any cost or goals with high tolerance. Such scenarios are not unique to CURROT. They occur under curriculum learning algorithms that overlook the constrained nature of a safety-critical setting. Therefore, to prioritize safety, a curriculum learning algorithm should have its objective aligned with the constrained RL problem. However, by construction, existing approaches suffer from misaligned objectives in constrained RL.

## 5 SAFE CURRICULUM GENERATION

We develop **Safe Curriculum Generation (SCG)**, a curriculum learning method that minimizes constraint violation regret and sample-efficiently learns an optimal policy (1). Algorithm 1 is a pseudocode for SCG. At curriculum iteration $k$, SCG samples contexts $\{\mathbf{x}_i\}_{i=0}^M$ from context distribution $\varrho_{k-1}$ (Line 5), and collects trajectories $\mathcal{D}_k = \{\tau_{\mathbf{x}_i}\}_{i=1}^M$, with transitions including the cost (Line 6). Then, a constrained RL algorithm updates policy $\pi_{k-1}$ (Line 7). Next, based on $\mathcal{D}_k$, the UPDATESUCCESSFULCONTEXTS() function determines *successful* contexts ($\mathcal{B}_+$) according to SCG's three phases (Line 8): 1) prioritizing safety, 2) prioritizing performance, and 3) safely approaching the target context distribution. Finally, SCG updates $V_r^\pi$ and $V_c^\pi$ based on $\mathcal{B}_+$ and $\mathcal{B}_-$ (Line 9) and generates the next context distribution $\varrho_k$ (Line 10) via

$$\Phi_{\text{SCG}}^\varphi(\pi_k, \varrho_+, \tilde{D}, \zeta) = \arg\min_\varrho \mathcal{W}_2(\varrho, \varphi) \quad \text{s.t.} \quad \varrho(\mathbf{x}) > 0 \Rightarrow V_r^{\pi_k}(\mathbf{x}) \geq \zeta, \forall \mathbf{x} \in \mathcal{X},$$

$$\varrho(\mathbf{x}) > 0 \Rightarrow V_c^{\pi_k}(\mathbf{x}) \leq \tilde{D}, \forall \mathbf{x} \in \mathcal{X},$$
$$\mathcal{W}_2(\varrho, \varrho_+) \leq \epsilon, \tag{4}$$

---

**Algorithm 1 S**afe **C**urriculum **G**eneration (SCG)

---

**Input**: Target and initial context distributions $\varphi$ and $\varrho_0$

**Parameters**: Safety threshold $D$, cost threshold $\tilde{D}$, performance threshold $\zeta$, Wasserstein distance bound $\epsilon$, number of curriculum iterations $K$, number of rollouts per iteration $M$, buffer size N

**Output**: Final policy $\pi_K$

1: Initialize policy $\pi_0$
2: $\mathcal{B}_-, \mathcal{B}_+ \leftarrow (), ()$         ▷ *initialize buffers of size N*
3: ISSAFE, ISPERF $\leftarrow$ False, False         ▷ *to search safe and performant contexts*
4: **for** $k = 1$ **to** $K$ **do**
5:      $\mathbf{x}_i \sim \varrho_{k-1}, i \in [M]$         ▷ *sample contexts*
6:      $\mathcal{D}_k = \{\boldsymbol{\tau}_{\mathbf{x}_i} = (\mathbf{s}_{i,t}, \mathbf{x}_i, \mathbf{a}_{i,t}, \mathbf{s}_{i,t+1}, r_{i,t}, c_{i,t})_{t=0}^T\}_{i=1}^M$    ▷ *collect rollouts via policy $\pi_{k-1}$*
7:      $\pi_k \leftarrow \Lambda(\mathcal{D}_k, \pi_{k-1}, D)$         ▷ *policy update via a constrained RL algorithm $\Lambda$*
8:      $\mathcal{B}_+, \mathcal{B}_-, \varrho_+,$ ISSAFE, ISPERF $\leftarrow$ UPDATESUCCESSFULCONTEXTS($\mathcal{B}_+, \mathcal{B}_-,$ ISSAFE, ISPERF, $\mathcal{D}_k$)
9:      Update value functions $V_r^{\pi_k}$ and $V_c^{\pi_k}$ with $\mathcal{B}_+$ and $\mathcal{B}_-$
10:      $\varrho_k \leftarrow \Phi_{\text{SCG}}^{\varphi}(\pi_k, \varrho_+, \tilde{D}, \zeta)$         ▷ *new context distribution (4)*

---

where, in contrast to CURROT, SCG imposes a constraint on $V_c^{\pi}$ to ensure that the support of the next distribution will be over low-cost contexts to minimize safety violations. Note that constrained RL algorithm $\Lambda$ utilizes safety threshold $D$ for the constraint on the expected cumulative cost (1), SCG uses cost threshold $\tilde{D}$ for a constraint on individual contexts under the support of $\varrho_k$. For the remainder of this section, we describe SCG's three phases (see Appendix C for more details). SCG does not sample contexts from $\varrho_+$ but uses it as a source distribution to approach $\varphi$ (4).

**1) Prioritizing safety.** Early in training, an RL agent likely collects high costs or low rewards during exploration. In a safety-critical setting, this period can rapidly increase constraint violation regret until the agent discovers how to behave safely. Therefore, SCG initially proposes *easy* contexts where the agent can behave safely without much exploration. To achieve that, UPDATESUCCESSFULCONTEXTS() labels safe contexts as *successful*. A context $\mathbf{x}$ is *safe* if the discounted cumulative cost $G_c(\boldsymbol{\tau}_{\mathbf{x}})$ is less than the median cost $C_{\text{med}}$ of $\mathcal{B}_+$. SCG updates $\mathcal{B}_+$ with safe contexts and generates $\varrho_+$, a Gaussian mixture model (GMM), via $\Xi_{\text{SAFE}}^{\text{INIT}}(\mathcal{B}_+)$ (5).

$$\Xi_{\text{SAFE}}^{\text{INIT}}(\mathcal{B}_+) = \sum_{\mathbf{x}_i \in \mathcal{B}_+} \omega_i \mathcal{N}(\mathbf{x}|\mathbf{x}_i, \sigma_{\text{SAFE},i}^2 \mathbf{I}), \tag{5}$$

where $\omega_i = \alpha_k \omega_i^c + (1 - \alpha_k)\omega_i^r$, $\omega_i^c \propto [C_{\text{med}} - G_c(\boldsymbol{\tau}_{\mathbf{x}_i})]_+$, $\omega_i^r \propto [G_r(\boldsymbol{\tau}_{\mathbf{x}_i}) - R_{\text{med}}]_+$, and $\sigma_{\text{SAFE},i} = \max\left\{\sigma_{\min}, 2\frac{G_c(\boldsymbol{\tau}_{\mathbf{x}_i}) - \tilde{D}}{C_{\max} - \tilde{D}}\right\}$. $C_{\max}$ is the maximum cost until curriculum iteration $k$ and $R_{\text{med}}$ is the median reward in $\mathcal{B}_+$. A weight $\omega_i$ of this Gaussian mixture model is a weighted average of $\omega_i^c$ and $\omega_i^r$ for context $\mathbf{x}_i$ with $\alpha_k \in [0, 1]$. The idea is to tune whether a context with low cost should have higher priority over a context with high reward. At the beginning of the search for safe contexts, SCG sets $\alpha_k$ to 1 and linearly anneals it to a lower pre-determined ratio unless that ratio is 1.

**2) Prioritizing performance.** Once $C_{\text{med}}$ is less than the cost threshold $\tilde{D}$, SCG focuses on performant contexts. A *performant* context has discounted cumulative reward $G_r(\boldsymbol{\tau}_{\mathbf{x}})$ greater than the median reward $R_{\text{med}}$ of $\mathcal{B}_+$. In the first two phases, $\mathcal{B}_+$ and $\mathcal{B}_-$ get updated cyclically. SCG generates $\varrho_+$ to be a GMM centered in contexts from $\mathcal{B}_+$. Performance prioritizing $\varrho_+$ differs from the previous one in terms of two factors: 1) SCG linearly anneals $\alpha_k$ to 0 unless it is 0. 2) The standard deviation parameter of this GMM is $\sigma_{\text{PERF},i} = \max\left\{\sigma_{\min}, 2\frac{\zeta - G_r(\boldsymbol{\tau}_{\mathbf{x}_i})}{\zeta - R_{\min}}\right\}$. Inspired by CURROT, SCG generates $\varrho_+$ in the first two phases as a GMM to allow for exploration in the context space.

**3) Safely approaching the target context distribution.** When $R_{\text{med}}$ exceeds $\zeta$, SCG moves to the final phase. Here, UPDATESUCCESSFULCONTEXTS() labels a context $\mathbf{x}$ as successful if the policy $\pi_{k-1}$ collects discounted cumulative reward greater than or equal to $\zeta$ and a discounted cumulative cost less than or equal to $\tilde{D}$. Similar to CURROT, to update a full success buffer, SCG generates a particle-based context distribution $\varrho_+(\mathbf{x}) = \frac{1}{|\mathcal{B}_+|}\sum_{\mathbf{x}_i \in \mathcal{B}_+} \delta_{\mathcal{B}_+}(\mathbf{x}_i)$, where $\delta_{\mathcal{B}_+}$ is a Dirac delta. Next, it replaces contexts in $\mathcal{B}_+$ with new ones from a distribution that minimizes the Wasserstein distance $\mathcal{W}_2(\varrho_+, \varphi)$. In contrast, $\mathcal{B}_-$ gets updated cyclically. SCG models $\varrho_+$ as a particle-based distribution since exploration is not as critical, and $\mathcal{B}_+$ well represents where the agent is safe and performant.

**Remark:** There are three novel algorithmic features that differentiate SCG from CURROT: 1) The cost constraint in (4) accounts for safety in any context. 2) Safety-prioritization via generating source distributions $\varrho_+$ over safe contexts (5) avoids high constraint violation regret. 3) Annealing mechanism to tune the prioritization ratio of safe or performant contexts in (5) and (6) allows for a smooth transition between safety and performance prioritization phases. Section 6.3 demonstrates how these features makes SCG a well-balanced, safe curriculum learning approach.

## 6   EMPIRICAL ANALYSIS

Our experiments in constrained RL domains investigate the following questions:

  6.1)  Can SCG obtain optimal policies while reducing safety violations?
  6.2)  Does SCG prioritize safe and performant contexts during training?
  6.3)  What is the contribution of SCG's components to its overall performance?
  6.4)  How do the final policies perform in each context of the target distribution?

To assess the quantitative questions, we consider **3 metrics**: 1) constraint violation regret (2) to evaluate safety during training, 2) expected discounted cumulative cost, and 3) expected success to assess safety and performance after training, respectively. For the qualitative questions, we visualize the evolution of curricula across the curriculum iterations.

We consider **3 constrained RL domains**: *safety-maze*, *safety-goal*, and *safety-push* (Figures 2 and 5a). In all domains, the agent aims to avoid hazards and reach a goal in the presence of misalignment phenomena. We study safety-maze to showcase that a simple modification to an existing domain (see Section 3.1) can trigger misaligned objectives. In comparison, safety-goal and safety-push are navigation tasks with realistic sensory observations in Safety-Gymnasium (Ji et al., 2023), a framework extensively used for constrained RL. In safety-goal, the agent has to navigate a car given high dimensional observations, namely, LIDAR outputs for the surrounding objects. Safety-push introduces (i) a larger observation space and more complicated dynamics, as the agent must push a box to a goal given additional LIDAR outputs, (ii) a layout favoring multi-modal distributions that draw goals on the left and right corridors, and (iii) a dead-end, trapping the agent and causing high costs. In all environments, the context determines the goal's location and tolerance. The target context distributions are uniform distributions over goals placed in the free space on the top row inside the walls/pillars. Appendix D.2 provides more details. We set the safety threshold $D = 0$ in our environments of interest, which necessitates safe paths to goals under $\varphi$. Nevertheless, SCG is not limited to such a constraint. The safety threshold in any other environment may be positive.

We compare SCG to **5 state-of-the-art curriculum learning methods**: CURROT (Klink et al., 2022), SPDL (Klink et al., 2021), PLR (Jiang et al., 2021b), GOALGAN (Florensa et al., 2018), and ALP-GMM (Portelas et al., 2020) (details in Appendix B). **3 baselines** provide perspective. To assess the impact of curricula, DEFAULT draws contexts from the target context distribution and, hence, does not generate curricula. To evaluate curricula that only consider the cost, CURROT4COST replaces the performance constraint of CURROT with a cost constraint $V_c^\pi(\mathbf{x}) \leq \tilde{D}$. To examine the need for an additional cost constraint in the curriculum update (4), NAIVESAFECURROT penalizes reward with cost and has an augmented performance constraint $(V_r^\pi(\mathbf{x}) - V_c^\pi(\mathbf{x}) \geq \zeta)$. All approaches use the same constrained RL algorithm, PPO-Lagrangian (Achiam & Amodei, 2019), to update the agent's policy. Appendix F.4 analyses other constrained RL algorithms with a DEFAULT curriculum.

### 6.1   CAN SCG OBTAIN OPTIMAL POLICIES WHILE REDUCING SAFETY VIOLATIONS?

Figures 4b and 4c show final policies' average cost and success in target contexts. while Fig. 4a illustrates the constraint violation regret (2) in training. Table 1 summarizes the results showing in which environments the most promising methods satisfied the criteria: 1) whether it yields an **optimal policy**, i.e., a policy achieving zero cost and the highest success rates in target contexts in the median. If so, it assesses 2) **safest training**, i.e., lowest constraint violation regret, and 3) **sample efficiency against DEFAULT** in terms of number of interactions to reach zero cost or the highest success rate (Appendix F.1 provides the corresponding learning curves).

**SCG yields zero cost and the highest success rate in all environments.** In detail, SCG and CURROT achieve the lowest costs, zero, in all domains, whereas the rest of the approaches achieve higher costs in multiple training runs in at least one domain. Note that the safety threshold $D$ (1) is zero

Table 1: Condensed analysis evaluates a method in safety-maze (M), safety-goal (G), and safety-push (P) based on three criteria. We strike through domains where a method violates a criterion, and '*' indicates failure in a run(s). We check the second and third criteria only if the first is satisfied.

| Method | Optimal policy | | | Safest training | | | Sample-efficient vs DEFAULT | | |
|---|---|---|---|---|---|---|---|---|---|
| SCG | M | G | P | M | G | P | M | G | P |
| CURROT | M | G̶ | P̶ | M̶ | | | M | | |
| NAIVESAFECURROT | M* | G̶ | P̶ | M | | | M | | |
| CURROT4COST | M̶ | G̶ | P̶ | | | | | | |

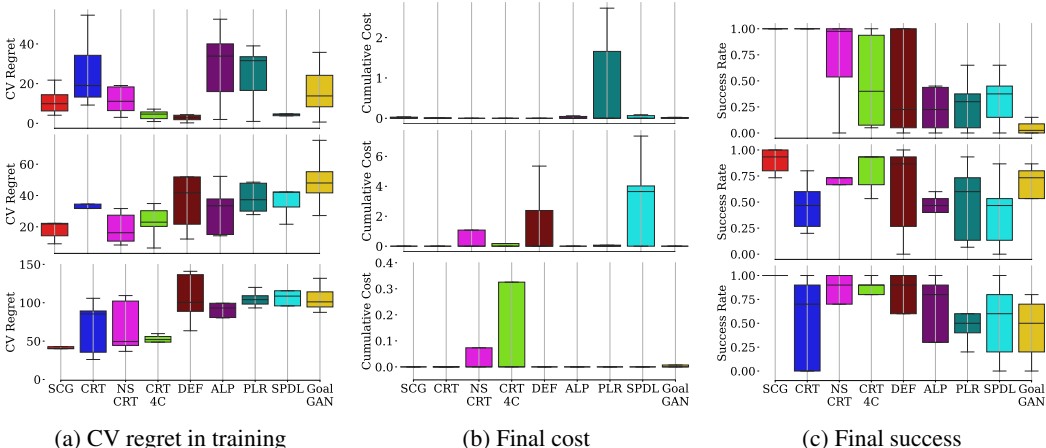

(a) CV regret in training      (b) Final cost      (c) Final success

Figure 4: Safety-maze (top), safety-goal (middle) and safety-push (bottom) from 10, 5, and 5 seeds, respectively. a) Constraint violation regret in training. b-c) Average discounted cumulative cost and success of the final policies in target contexts, respectively. Box plots show the min, first quartile, median, third quartile, and max, from bottom to top. We use CRT for CURROT, NSCRT for NAIVESAFECURROT, DEF for DEFAULT, ALP for ALP-GMM, and GGAN for GOALGAN.

for all domains; hence any method yielding non-zero cost is considered unsafe. Although CURROT attains 100% success in safety-maze, it fails to compete in other domains. NAIVESAFECURROT and CURROT4COST either cannot achieve the highest median success or have high variance among training runs. In all environments, at least one curriculum learning method outperforms DEFAULT, no curriculum baseline, in terms of final cost and success, evidencing that automated curricula can boost final performance/safety. However, ALP-GMM, PLR, SPDL, and GOALGAN fall behind.

**In all domains, SCG reaches the lowest constraint violation regret among methods that achieve zero cost and highest success rates.** Although some seem safer during training, they fail to learn optimal policies. For example, in safety-maze (top row), NAIVESAFECURROT and CURROT4COST yield similar or lower constraint violation regret than SCG, but they have varying success rates. CURROT achieves zero cost and 100% success, yet it yields high constraint violation regret in safety-maze, as it suffers from misaligned objectives. One can make similar arguments for the other domains. On a side note, DEFAULT either attains the lowest (safety-maze) or one of the highest constraint violation regrets while failing to learn optimal policies consistently. This may be because DEFAULT either causes the agent to be conservative yet suboptimal or to collect high costs.

**SCG consistently yields optimal policies.** In safety-maze, CURROT and NAIVESAFECURROT achieve the highest success rates in the median, yet they fail to learn optimal policies in other domains. Both are sample-efficient compared to DEFAULT, but CURROT causes higher constraint violation regret so it is not the safest. CURROT4COST and the rest of the approaches learn suboptimal policies. Thus, we do not evaluate them in terms of safety or sample efficiency.

## 6.2 DOES SCG PRIORITIZE SAFE AND PERFORMANT CONTEXTS DURING TRAINING?

Fig. 3 and Fig. 5 show CURROT's and SCG's curricula in different epochs. **Overall, SCG prioritizes safe and performant contexts as it resolves the misalignment between curriculum learning and constrained RL.** In safety-maze, SCG initially prioritizes goals with high tolerance over the

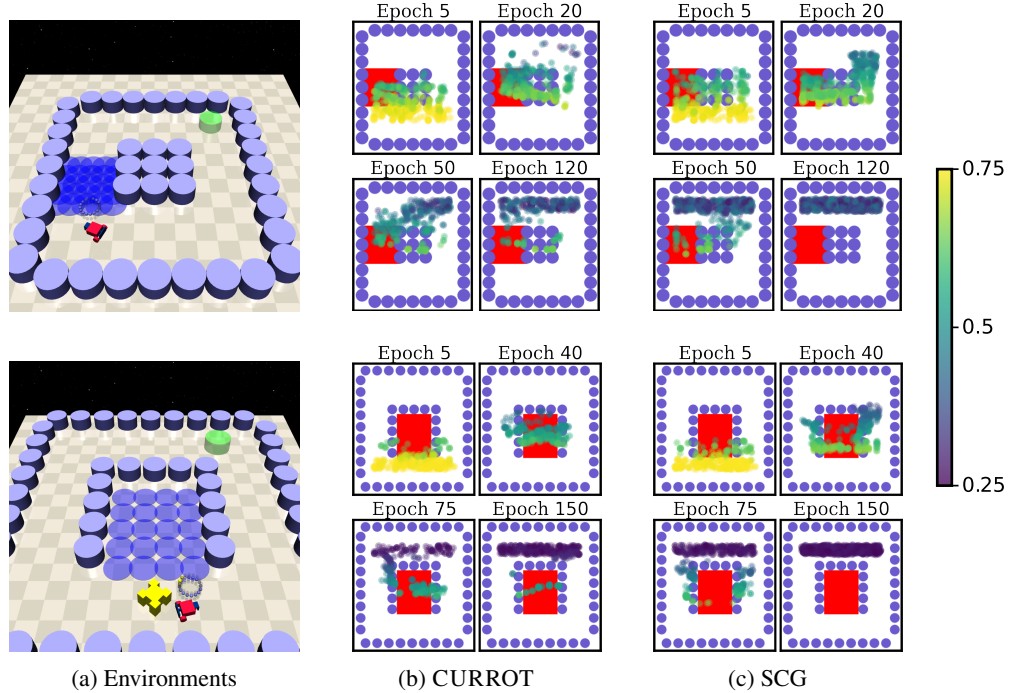

|         |         |         |
| ------- | ------- | ------- |
| (a) Environments | (b) CURROT | (c) SCG |

Figure 5: SCG and CURROT'curricula in safety-goal (top) and safety-push (bottom). The first column shows environment snapshots. The second and third columns demonstrate the curricula generated by CURROT and SCG, respectively. In safety-goal and safety-push, we draw contexts from distributions $\varrho_k$ at iterations/epochs $k \in \{5, 20, 50, 120\}$ and $k \in \{5, 40, 75, 150\}$, respectively.

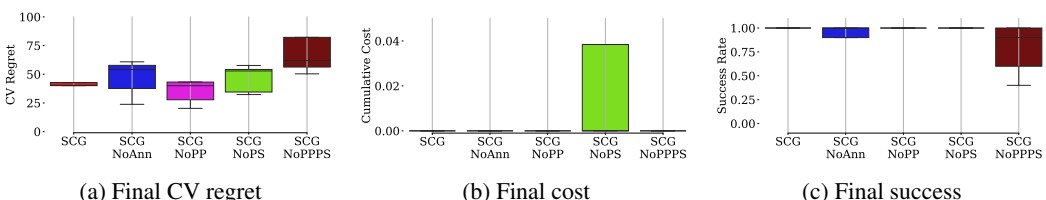

|         |         |         |
| ------- | ------- | ------- |
| (a) Final CV regret | (b) Final cost | (c) Final success |

Figure 6: Ablation study in safety-push to investigate the impact of SCG's main components: Phase 1, which prioritizes safety (PS), Phase 2, which prioritizes performance (PP), and annealing (Ann).

bottom white row since they are *easy*. Eventually, SCG moves goals over the right white column with decreasing tolerances. The goals SCG places in red and black areas have a high tolerance and can be reached outside both regions. CURROT moves contexts through the hazardous area, causing safety violations. Similarly, in safety-goal SCG prioritizes contexts on the right side, as they allow the agent to learn how to reach the target contexts by avoiding the hazards. In contrast, safety-push's map favors multi-modal distributions with the hazards in the middle. SCG recognizes that and prioritizes contexts on the left and right corridors. However, CURROT moves contexts through the hazards, causing high constraint violation regret (see Fig. 4a). See Appendix F.5 for the progression of curricula generated by the rest of the approaches.

## 6.3 WHAT IS THE CONTRIBUTION OF SCG'S COMPONENTS TO ITS OVERALL PERFORMANCE?

Fig. 6 demonstrates the results in safety-push obtained by four variations of SCG: 1) SCG-NOANN does not do annealing, 2) SCG-NOPP prioritizes safety first and then moves on to Phase 3 by skipping performance prioritization, 3) SCG-NOPS begins training at Phase 2, hence does not prioritize safety initially, and lastly 4) SCG-NOPPPS starts the training at Phase 3, hence does not prioritize safety nor performance individually.

SCG and SCG-NOPP yield the lowest constraint violation regret during training (in the median, see Fig. 6a), whereas SCG-NOPS and SCG-NOANN closely follow, and SCG-NOPPPS yields

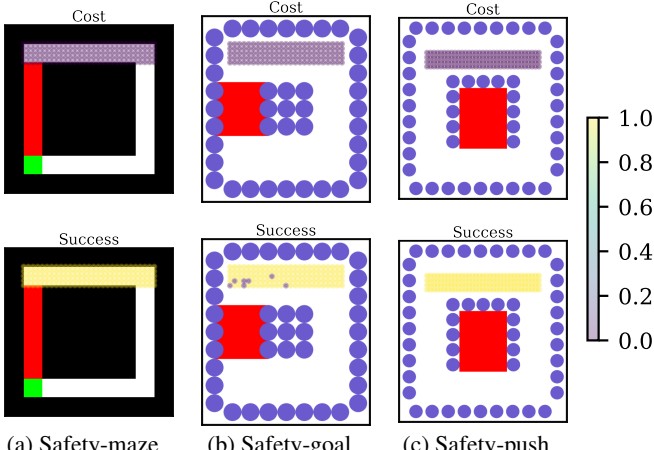

(a) Safety-maze     (b) Safety-goal     (c) Safety-push

Figure 7: In-distribution generalization capability of SCG. A marker's color and position demonstrate the median cost/success and the goal location. We set the tolerance to the minimum value. We divide the supports of target context distributions into grids and pick the goal locations in their centers.

the highest CV regret. In terms of final performance, all except SCG-NoPPPS reach the highest success rate in median (see Fig. 6c). Although all achieve zero cost in target contexts in the median, SCG-NoPS yields unsafe policies in multiple runs at test-time (see Fig. 6b). Also, SCG is the most sample efficient, followed by SCG-NoPP (see Appendix F.2). Overall, PS provides lower constraint violation regret, yet the learning progress slows down when not combined with PP and annealing. In addition, Appendix F.3 provides an ablation study for several SCG parameters.

### 6.4 How do the final policies perform in each context under the target distribution?

The CCRL problem (1) averages the expected return and cost across the target contexts according to the target distribution ($\varphi$). Thus, we evaluate the policies learned in the individual contexts in terms of success and cost (Fig. 7). We set the goal tolerance of these contexts to the minimum value to escalate difficulty. The results evidence that SCG yields policies that act safely, that is, they receive zero cost, and succeed in all target contexts except for some goals placed just above the hazardous area in safety-goal. In summary, although we consider an average across the contexts, the policies learned are still safe and performant in individual contexts.

## 7 Conclusion

In this work, we study safe automated curriculum generation in multi-task cost-constrained settings with distributions over target tasks. We propose a safe curriculum generation approach (SCG) developed for constrained RL to minimize constraint violation regret and accelerate learning. SCG initially prioritizes tasks with low costs over high-reward ones, to ensure that the agent learns a policy that satisfies the cost constraint. Next, SCG proposes tasks where the agent can collect high rewards. Finally, SCG takes safety and performance jointly into account. Our empirical evaluation evidences that state-of-the-art curriculum learning approaches fail to learn optimal behavior in a safe and stable way, as these approaches suffer from misaligned objectives with constrained RL. In contrast, SCG obtains optimal behavior with the lowest constraint violation regret in domains with low or high dimensional state spaces and complicated dynamics and task structures.

**Limitations and future work.** While SCG can reduce the constraint violation regret while preserving the benefits of curriculum learning in boosting learning speed, it does not provide any guarantees for the constraint violation regret that is achieved at the end of the training. Combining SCG with constrained RL algorithms that guarantee safety could mitigate this issue while keeping a good learning speed. Furthermore, SCG inherits a dependency on the initial context distribution from automated curriculum generation. Thus, if the support of the initial context distribution is unsafe, it might take multiple iterations to find safe contexts.

## REPRODUCIBILITY STATEMENT

For all the hyperparameters and detailed settings of the experiments, please refer to Appendix D. We put the core code of SCG in the supplementary details. The code includes instructions to install necessary software, reproduce the experiments and target contexts where we evaluate trained policies to obtain the experimental results.

## ACKNOWLEDGMENTS

This work is supported by the Office of Naval Research (ONR) under grant numbers N00014-24-1-2797 and N00014-22-1-2254, and the European Research Council (ERC) under the starting grant 101077178 (DEUCE).

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

# A    NOMENCLATURE

| | |
|---|---|
| $\mathbf{x}$ | Context |
| $\mathcal{M}$ | Contextual constrained MDP |
| $\mathcal{S}, \mathcal{A}, \mathcal{X}$ | State, action, and context spaces |
| $\mathsf{M}$ | Mapping from contexts to constrained MDPs |
| $\mathbf{s}, \mathbf{a}, r, c$ | State, action, reward, cost |
| $p_{\mathbf{x}}, r_{\mathbf{x}}, c_{\mathbf{x}}$ | Transition, reward and cost functions |
| $p_{0,\mathbf{x}}$ | Initial state distribution |
| $\pi$ | Policy |
| $\gamma$ | Discount factor |
| $D$ | Safety threshold |
| $V_r^{\pi}, V_c^{\pi}$ | Value function for reward and cost |
| $\varrho, \varphi$ | Context distribution and target context distribution |
| $J(\pi, \varphi)$ | CCRL objective function |
| $\boldsymbol{\tau}_{\mathbf{x}}$ | Trajectory in context $\mathbf{x}$ |
| $G_r, G_c$ | Discounted cumulative reward and cost in a trajectory |
| $\text{Reg}^{tr}$ | Constraint violation regret during training |
| $\Lambda$ | CCRL algorithm |
| $L$ | Number of episodes during CCRL training |
| $l, T$ | A CCRL episode and its length |
| $\tilde{D}, \zeta, \epsilon$ | Cost, performance and divergence thresholds in SCG |
| $\mathcal{D}$ | Trajectory set |
| $\Phi_{\text{SCG}}^{\varphi}$ | SCG function |
| $\omega$ | GMM weight |
| $\sigma_{\text{SAFE},i}, \sigma_{\text{PERF},i}$ | GMM's standard deviations for safe/performant context $\mathbf{x}_i$ |
| $\sigma_{\min}$ | Minimum standard deviation in GMM |
| $\omega^r, \omega^c$ | GMM weights for discounted cumulative reward and cost |
| $\alpha_k$ | GMM weight ratio |
| $C_{\text{med}}, C_{\text{max}}$ | Median and maximum cost in buffer |
| $R_{\text{med}}, R_{\min}$ | Median and minimum reward in buffer |
| $\Xi_{\text{SAFE}}^{\text{INIT}}, \Xi_{\text{PERF}}^{\text{INIT}}, \Xi^{\text{MAIN}}$ | Phase 1, 2 and 3 update functions in SCG |
| $\mathcal{B}_+, \mathcal{B}_-, \mathcal{B}_+^{\text{TEMP}}, N$ | Success, failure and temporary success buffers and its size |
| $\varrho_+$ | Distribution over successful contexts |
| UPDATESUCCESSFULCONTEXTS() | Update function for successful contexts |
| $k, K$ | Curriculum iteration and total number of iterations |
| $M$ | Number of rollouts per curriculum iteration |

## B    Automated Curriculum Generation Algorithms

In this section, we provide short descriptions of the state-of-the-art curriculum learning methods evaluated in the experiments.

- CURROT (Klink et al., 2022): We propose SCG based on *Curriculum RL via Constrained Optimal Transport*, which we describe and discuss in Sections 4.1 and 4.2.

- SPDL (Klink et al., 2021): *Self-paced Deep Reinforcement Learning* formulates the automated curriculum generation problem similarly to CURROT, except that SPDL generates context distributions that minimize the KL divergence to the target context distribution. The constraints in the optimization problem solved in SPDL are on minimum expected discounted cumulative reward and maximum KL divergence to the previous context distribution. SPDL does not include an initial search procedure and generates context distributions as Gaussian distributions. SPDL generates parametric context distributions and does not provide the robustness that CURROT achieves via a performance constraint on individual constraints. Therefore, we can highlight why CURROT is better for building SCG.

- PLR (Jiang et al., 2021b): *Prioritized Level Replay* is a curriculum learning method developed for procedural context generation environments, where a *level* corresponds to a task, i.e., an environment instance. PLR prioritizes levels that have a high average magnitude of generalized advantage estimate (Schulman et al., 2016), namely, the discounted sum of temporal-difference errors. PLR is a popular curriculum learning approach, yet it is unsuitable for continuous context spaces, as it assumes that contexts are discrete and unidimensional. In addition, PLR assumes uniform target context distributions.

- GOALGAN (Florensa et al., 2018): *Goal Generative Adversarial Network* is a curriculum learning approach developed for goal-conditioned RL. GOALGAN trains a goal discriminator to classify goals that are at the intermediate difficulty for the policy of the RL agent and a goal generator to generate goals at that difficulty to boost learning performance. GOALGAN is among the first automated curriculum generation approaches that pose curriculum as a sequence of distributions. Yet, GOALGAN does not interpolate as in CURROT and SPDL. Similar to PLR, GOALGAN assumes uniform target context distributions.

- ALP-GMM (Portelas et al., 2020): *Absolute Learning Progress with Gaussian Mixture Models* uses the absolute learning progress of a task to measure whether a task would improve the learning process of an RL agent. ALP-GMM learns a Gaussian mixture model over the absolute learning progress where a multi-armed bandit samples a Gaussian as an arm based on its utility, which is the absolute learning progress. The Gaussian distribution that the arm corresponds to draws a task, namely, the context in our setting. ALP-GMM also generates a sequence of context distributions, but similar to GOALGAN, it does not interpolate them. In addition, it also assumes that the target context distribution is uniform.

## C    Details of SCG

To support Section 5, here we provide a closer look into how the UPDATESUCCESSFULCONTEXTS() function in SCG works (See Algorithm 2 for a pseudocode).

**1) Prioritizing safety.** Initially, SCG sets FOUNDSAFEXS and FOUNDPERFXS to false to enable the UPDATESUCCESSFULCONTEXTS() function to search for safe contexts first. Lines 2-3 indicate that a successful context in this phase yields a discounted cumulative cost less than or equal to the median cost $C_{\text{med}}$ in success buffer $\mathcal{B}_+$. Cyclically, $\mathcal{B}_+$ gets updated with the successful contexts in trajectory set $\mathcal{D}_k$, as $\mathcal{B}_-$ gets updated with the rest of the contexts. Then, UPDATESUCCESSFULCONTEXTS() generates $\varrho_+$ as a Gaussian mixture model using $\mathcal{B}_+$ (Line 4) (see (5) in Section 5). SCG searches for such safe contexts until $C_{\text{med}}$ is less than or equal to cost threshold $\tilde{D}$ (Line 5).

**2) Prioritizing performance.**    Once    the    contexts    in    $\mathcal{D}_k$    satisfy    the    safety    condition, UPDATESUCCESSFULCONTEXTS() switches its focus to finding performant contexts. Similarly, SCG begins by updating $\mathcal{B}_+$ with contexts where the discounted cumulative reward is greater than or equal to the performance threshold $\zeta$ (Lines 7-8). Then, SCG uses $\Xi_{\text{PERF}}^{\text{INIT}}(\mathcal{B}_+)$ to generate $\varrho_+$, which

---

**Algorithm 2** UPDATESUCCESSFULCONTEXTS()

---

**Input**: $\mathcal{B}_+, \mathcal{B}_-, \text{ISSAFE}, \text{ISPERF}, \mathcal{D}_k$
**Parameters**: Cost threshold $\tilde{D}$, performance threshold $\zeta$
**Output**: $\mathcal{B}_+, \mathcal{B}_-, \varrho_+, \mathcal{D}_k, \text{ISSAFE}, \text{ISPERF}$

1:  **if** not FOUNDSAFEXS **then**
2:      Add $\{\mathbf{x}_i | G_c(\boldsymbol{\tau}_{\mathbf{x}_i}) > C_{\text{med}})\}$ to $\mathcal{B}_-$
3:      Add $\{\mathbf{x}_i | G_c(\boldsymbol{\tau}_{\mathbf{x}_i}) \leq C_{\text{med}}\}$ to $\mathcal{B}_+$
4:      $\varrho_+ \leftarrow \Xi_{\text{SAFE}}^{\text{INIT}}(\mathcal{B}_+)$                                    ▷ *prioritize safety*
5:      FOUNDSAFEXS $\leftarrow C_{\text{med}} \leq \tilde{D}$
6:  **else if** not FOUNDPERFXS **then**
7:      Add $\{\mathbf{x}_i | G_r(\boldsymbol{\tau}_{\mathbf{x}_i}) < R_{\text{med}}\}$ to $\mathcal{B}_-$
8:      Add $\{\mathbf{x}_i | G_r(\boldsymbol{\tau}_{\mathbf{x}_i}) \geq R_{\text{med}}\}$ to $\mathcal{B}_+$
9:      $\varrho_+ \leftarrow \Xi_{\text{PERF}}^{\text{INIT}}(\mathcal{B}_+)$                                    ▷ *prioritize performance*
10:     FOUNDPERFXS $\leftarrow R_{\text{med}} \geq \zeta$
11: **else**
12:     Add $\{\mathbf{x}_i | G_r(\boldsymbol{\tau}_{\mathbf{x}_i}) < R_{\text{med}} \text{ or } G_c(\boldsymbol{\tau}_{\mathbf{x}_i}) > C_{\text{med}}\}$ to $\mathcal{B}_-$
13:     $\mathcal{B}_+^{\text{TEMP}} \leftarrow \{\mathbf{x}_i | G_r(\boldsymbol{\tau}_{\mathbf{x}_i}) \geq R_{\text{med}} \text{ and } G_c(\boldsymbol{\tau}_{\mathbf{x}_i}) \leq C_{\text{med}}\}$
14:     $\mathcal{B}_+, \varrho_+ \leftarrow \Xi^{\text{MAIN}}(\mathcal{B}_+^{\text{TEMP}}, \mathcal{B}_+, \varphi)$                                    ▷ *main phase*
15: **return** $\mathcal{B}_+, \mathcal{B}_-, \varrho_+, \text{ISSAFE}, \text{ISPERF}, \mathcal{D}_k$

---

differs from $\Xi_{\text{SAFE}}^{\text{INIT}}(\mathcal{B}_+)$ in terms of GMM weights $\omega^r$ and standard deviation $\sigma_{\text{PERF},i}$ (Line 9).

$$\Xi_{\text{PERF}}^{\text{INIT}}(\mathcal{B}_+) = \sum_{\mathbf{x}_i \in \mathcal{B}_+} \omega_i \mathcal{N}(\mathbf{x}|\mathbf{x}_i, \sigma_{\text{SAFE},i}^2 \mathbf{I}), \quad \text{where} \quad \omega_i = \alpha_k \omega_i^c + (1 - \alpha_k)\omega_i^r, \tag{6}$$

$$\omega_i^c \propto \max\{0, C_{\text{med}} - G_c(\boldsymbol{\tau}_{\mathbf{x}_i})\}, \quad \omega_i^r \propto \max\{0, G_r(\boldsymbol{\tau}_{\mathbf{x}_i}) - R_{\text{med}}\}, \quad \text{and}$$

$$\sigma_{\text{PERF},i} = \max\left\{\sigma_{\min}, 2\frac{\zeta - G_r(\boldsymbol{\tau}_{\mathbf{x}_i})}{\zeta - R_{\min}}\right\}.$$

Note that $R_{\min}$ is the minimum reward until curriculum iteration $k$. SCG prioritizes performant contexts until $R_{\text{med}}$ is greater than or equal to $\zeta$ (Line 10). During the initial search, SCG updates $\mathcal{B}_+$ and $\mathcal{B}_-$ in a cyclic fashion.

**3) Safely approaching the target context distribution.** Section 5 already provides information about how the last phase of SCG works. This phase operates similarly to the main phase of CURROT. For a detailed description, we refer the reader to Klink et al. (2022).

## D    EXPERIMENTAL DETAILS

We discuss the process of hyperparameter selection for the curriculum learning approaches evaluated in this work and additional details about the constrained RL environments in the experiments.

### D.1    ALGORITHM HYPERPARAMETERS

SCG has five main parameters: performance threshold $\zeta$, cost threshold $\tilde{D}$, Wasserstein distance threshold $\epsilon$, number of curriculum iterations $K$ and number of rollouts per curriculum updates $M$. CURROT and NAIVESAFECURROT share the same parameters except the cost threshold $\tilde{D}$, whereas CURROT4COST shares all except the performance threshold $\zeta$. We chose $\zeta$ to be approximately the midpoint between the minimum and maximum possible discounted cumulative reward or success rate. To select the Wasserstein distance threshold $\epsilon$, we ran a grid search over $\{0.25, 0.5\}$ for safety-goal, and over $\{1.0, 1.25\}$ for safety-maze. For the number of rollouts per curriculum updates $M$, we ran grid searches over $\{20, 40\}$ for all settings. Although SPDL shares $\zeta$, $K$, and $M$, it has a KL divergence threshold $\epsilon_{\text{KL}}$, for which we ran a grid search over $\{0.25, 0.5\}$ in all environments. Table 2 provides all parameter values. The initial search procedure in SCG has three hyperparameters: ratio $\alpha_k$ and minimum standard deviation $\sigma_{\min}$ for the Gaussian mixture model, and lastly the number steps to anneal $\alpha_k$. In all settings, we examined linearly annealing $\alpha_k$ from 1 to 0.75 for searching safe contexts and 0.75 to 1.0 for performant contexts in 10 or 20

Table 2: Parameters used for SCG, CURROT, NAIVESAFECURROT, and CURROT4COST.

| Environment | $\zeta$ | $\tilde{D}$ | $\epsilon_{KL}$ | $\epsilon$ | $K$ | $M$ |
|---|---|---|---|---|---|---|
| Safety-maze | 0.6 | 0.25 | 0.25 | 1.25 | 500 | 40 |
| Safety-goal | 0.6 | 1 | 0.25 | 0.5 | 150 | 20 |
| Safety-passage | 0.6 | 1 | 0.25 | 0.5 | 200 | 20 |
| Safety-push | 0.6 | 1 | 0.25 | 0.25 | 300 | 40 |

Table 3: Selected values for parameters of PLR, GOALGAN and ALP-GMM

| Environment | $\rho$ | $\beta$ | $p$ | $\delta_{noise}$ | $n_{rollout}^{GG}$ | $p_{success}$ | $p_{rand}$ | $n_{rollout}^{AG}$ | $s_{buffer}$ |
|---|---|---|---|---|---|---|---|---|---|
| Safety-maze | 0.45 | 0.15 | 100 | 0.1 | 200 | 0.2 | 0.2 | 200 | 500 |
| Safety-goal | 0.45 | 0.15 | 100 | 0.1 | 200 | 0.2 | 0.2 | 200 | 500 |
| Safety-passage | 0.45 | 0.15 | 100 | 0.1 | 200 | 0.2 | 0.2 | 200 | 500 |
| Safety-push | 0.45 | 0.15 | 100 | 0.1 | 200 | 0.2 | 0.2 | 200 | 500 |

curriculum iterations. We did not anneal $\alpha_k$ for safety-maze and safety-goal, but annealed $\alpha_k$ in 10 iterations for safety-push and safety-passage. We fix $\sigma_{min} = 0.001$ as in CURROT.

As parameters to tune, PLR has the score temperature $\beta$, the staleness coefficient $\rho$, and the replay probability $p$. We ran a grid search over $(\rho, \beta, p) \in \{0.15, 0.45\} \times \{0.15, 0.45\} \times \{0.55, 0.85\}$. GOALGAN has three parameters: the number of rollouts between curriculum updates $n_{rollout}^{GG}$, the random noise on drawn contexts $\delta_{noise}$, and the percentage of contexts to draw from the success buffer $p_{success}$. We ran a grid search over $(\delta_{noise}, n_{rollout}^{GG}, p_{success}) \in \{0.05, 0.1\} \times \{100, 200\} \times \{0.1, 0.2\}$. ALP-GMM has three parameters: the buffer size $s_{buffer}$, the number of rollouts between curriculum updates $n_{rollout}^{AG}$, and the probability of randomly sampling contexts $p_{rand}$. We ran a grid search over $(p_{rand}, n_{rollout}^{AG}, s_{buffer}) \in \{0.1, 0.2\} \times \{50, 100\} \times \{500, 1000\}$. Table 3 shows the final parameter used for PLR, GOALGAN, and ALP-GMM.

### D.2 ENVIRONMENT DESCRIPTIONS

**Safety-maze environment.** Inspired by the maze environment (Klink et al., 2022), we design safety-maze, where the agent receives rewards from of -1 until it reaches the goal, and costs of 0.25 when it enters the hazardous area (Figure 2). The context space $\mathcal{X} = [-9, 9] \times [-9, 9] \times [0.25, 5.0]$ is over $x$ and $y$ positions and tolerances of the goal, respectively. We train a constrained RL agent using the PPO-Lagrangian algorithm Achiam & Amodei (2019). The implementation we integrate into our codebase is from OmniSafe Ji et al. (2024). The parameters of the PPO-Lagrangian are fixed to their default values in OmniSafe, except the number of steps to update the policy is 4000 and the number of iterations to update the policy is 12.

**Safety-goal environment.** To evaluate SCG in more complex dynamics, we create an environment with a high-dimensional state space in Safety-Gymnasium Ji et al. (2023). We use pillars, purple columns, and hazards, blue circles, as objects with which the agent, a car, interacts in the environment (see Figure 5a). The rewards and costs come from the safety-gymnasium implementation. The context space $\mathcal{X} = [-1.5, 1.5] \times [-1.5, 1.5] \times [0.25, 0.75]$ is goal positions and tolerances, respectively. Similar to safety-maze, we only change the parameters of PPO-Lagrangian in OmniSafe by setting the number of steps and iterations to update the policy to 10000 and 15, respectively.

**Safety-passage environment.** Similar to safety-goal, we create an environment with a high-dimensional state space in Safety-Gymnasium Ji et al. (2023). We use pillars, purple columns, and hazards, blue circles, as objects with which the agent, a car, interacts in the environment (see Figure 8). The rewards and costs come from the safety-gymnasium implementation. The context space $\mathcal{X} = [-1.5, 1.5] \times [-1.5, 1.5] \times [0.25, 0.75]$ is goal positions and tolerances, respectively. Similar to safety-maze, we change the parameters of the PPO-Lagrangian implementation in OmniSafe by setting the number of steps and iterations to update the policy to 10000 and 15, respectively. In addition, the architecture of a policy is a two-layered perception with 256 neurons in each.

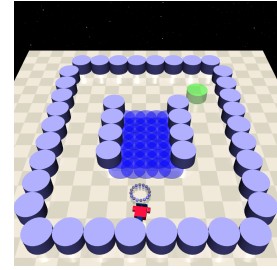

Figure 8: Safety-passage.

**Safety-push environment.** In addition to the objects in safety-goal, safety-push includes a box for the agent to push to a goal while avoiding hazards (see Figure 5a). The rewards and costs come from the safety-gymnasium implementation. Similar to safety-maze, we only change the parameters of the PPO-Lagrangian implementation in OmniSafe by setting the number of steps and iterations to update the policy to 10000 and 15, respectively, and the batch size to 256. The context space $\mathcal{X} = [-1.5, 1.5] \times [-1.5, 1.5] \times [0.25, 0.75]$ is goal positions and tolerances, respectively. As in safety-passage, the architecture of a policy is a two-layered perception with 256 neurons in each.

## E    COMPUTATION RESOURCES

We run our experiments on a cluster with NVIDIA RTX A5000 GPUs and an Intel(R) Xeon(R) Gold 6226R CPU @ 2.90GHz. We utilize Omnisafe Ji et al. (2024) as our RL framework, which uses 16 torch threads and no parallel environments in our experiments. A training run in the experimented environments approximately takes the following number of hours: (i) safety-maze: 5.5 hours (6000 gradient steps in 2 million interactions), (ii) safety-goal: 7 hours (2250 gradient steps in 1.5 million interactions), (iii) safety-passage: 9 hours (3000 gradient steps in 2 million interactions), and (iv) safety-push: 12 hours (4500 gradient steps in 3 million interactions).

## F    DETAILED ANALYSIS OF RESULTS

### F.1    QUANTITATIVE RESULTS

In this section, we present figures that complement Fig. 4, which demonstrate the constraint violation regret at the end of the training and the average cost and success of final policies in target contexts.

#### F.1.1    SAFETY-MAZE

Figure 9 demonstrates additional plots that provide detailed information about safety and performance during and after training. We observe that SCG, CURROT, NAIVESAFECURROT, CURROT4COST, and DEFAULT achieve optimal behavior in at least one run out of 10. However, SCG and CURROT consistently get optimal policies with converged constraint violation regret during training and with respect to the target context distribution. We highlight that, as SCG paces the curriculum according to how safely the agent behaves, its constraint violation regret in $\varphi$ converges the last. Nevertheless, it achieves the lowest constraint violation regret in training out of all approaches that more or so reliably learn an optimal policy. ALP-GMM, PLR, SPDL, and GOALGAN achieve similar success rates throughout the training and they all fail in target contexts. However, the constraint violation regret of ALP-GMM and PLR in training increases very rapidly, with GOALGAN following behind. In contrast, the rest of the approaches have converged constraint violation regret in training.

#### F.1.2    SAFETY-GOAL

The results in Figure 10 demonstrate that SCG generates curricula that achieve the highest success rates and the lowest costs during training time and when deployment after. CURROT4COST and DEFAULT can achieve similar success rates, but not as reliably, in target contexts. DEFAULT has its best performance out of all three constrained environments we study because safety-goal has a dense reward function, which eases learning without a curriculum. NAIVESAFECURROT and CURROT4COST also yield as low constraint violation regret as SCG has at the final iteration of the training. However, NAIVESAFECURROT is less stable at learning policies that accomplish the task both during and after training. CURROT, ALP-GMM, SPDL, and GOALGAN follow NAIVESAFECURROT in terms of success in target contexts. An important interpretation to make in all three settings is that, in contrast to learning directly in target contexts, a curriculum learning approach can cause an agent to behave unsafely if the approach does not consider the constrained nature of the task.

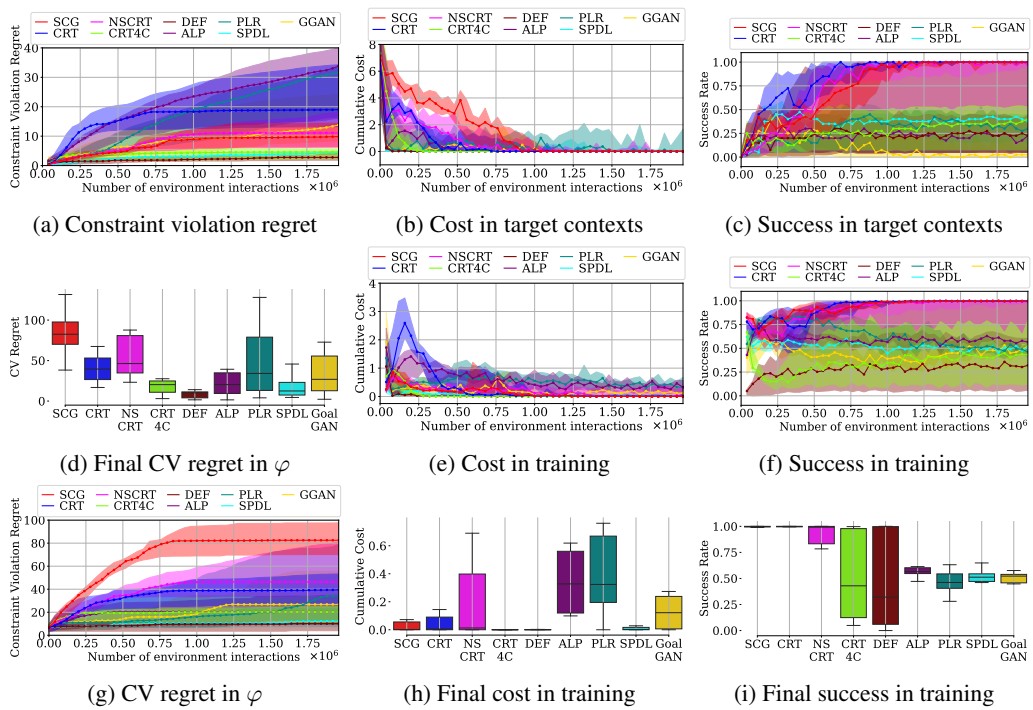

Figure 9: Safety-maze results in 10 seeds: a) Evolution of constraint violation regret during. b) Progression of expected discounted cumulative cost in target contexts. c) Progression of expected success rate in contexts drawn from the target context distribution. d) Constraint violation regret with respect to the target context distribution at the final curriculum iteration. e) Progression of expected discounted cumulative cost in contexts sampled during training. f) Progression of expected success rate in contexts sampled during training. g) Evolution of constraint violation regret with respect to the target context distribution. h) Expected discounted cumulative cost of the final policies in contexts sampled during training. i) Expected success rate of the final policies in contexts sampled during training.

### F.1.3 SAFETY-PASSAGE

Similar to safety-goal, the results in Figure 11 demonstrate that SCG generates curricula that achieve the highest success rates and the lowest costs during training time and when deployment after. In target contexts, NAIVESAFECURROT and DEFAULT can achieve similar success rates but less reliably. CURROT and CURROT4COST can get close but still fail in some runs. ALP-GMM, SPDL, PLR, and GOALGAN fall behind in safety and performance by a big margin.

### F.1.4 SAFETY-PUSH

Safety-push results in Figure 12 support our observations in the rest of the environments. SCG successfully aligns the objectives of constrained RL and curriculum learning, thus yielding the lowest constraint violation regret while achieving the lowest cost and highest success in target contexts. Although DEFAULT, CURROT and its naively modified safe versions can get close to SCG in terms of final success, they are not as safe during training and do not consistently yield safe agents at the end. As in the other environments, PLR, SPDL, and GOALGAN fall behind in terms of safety and performance, whereas ALP-GMM seems to be as performant as CURROT, but not as safe.

### F.2 ABLATION STUDY FOR SCG COMPONENTS

Fig. 13 provide progression plots for average cost, success, and return in contexts generated during training and sampled from the target context distribution. As mentioned in the main document, SCG is the first to reach 100% success in target contexts, where SCG-NOPP closely follows. Although

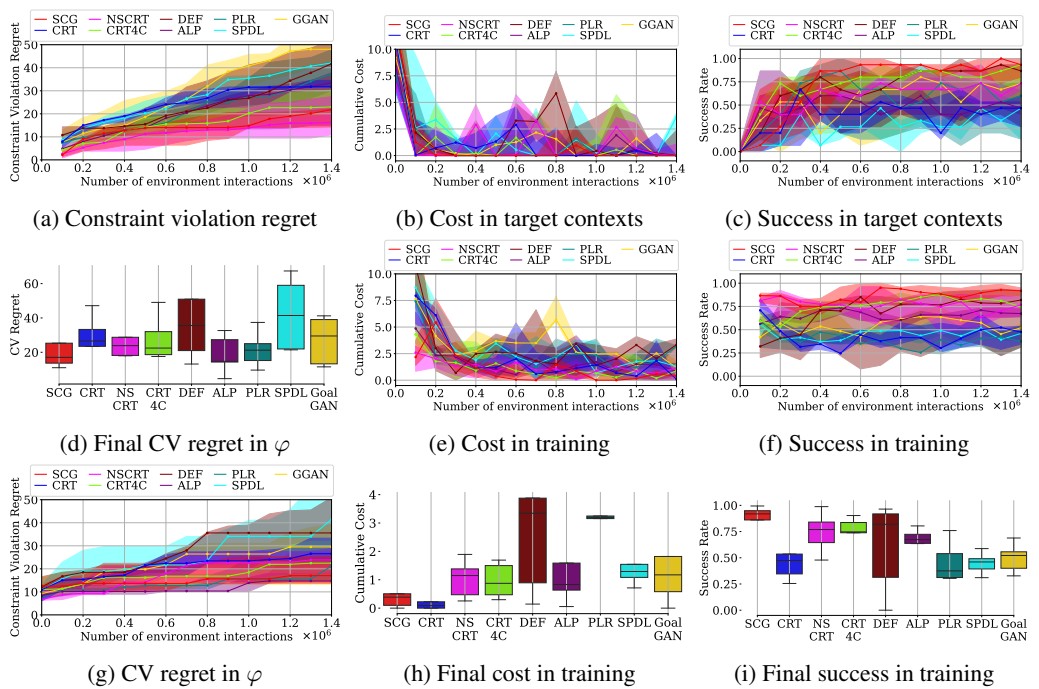

Figure 10: Safety-goal results from runs in 5 seeds: a) Evolution of constraint violation regret during. b) Progression of expected discounted cumulative cost in contexts drawn from the target context distribution. c) Progression of expected success rate in contexts drawn from the target context distribution. d) Constraint violation regret with respect to the target context distribution at the final curriculum iteration. e) Progression of expected discounted cumulative cost in contexts sampled during training. f) Progression of expected success rate in contexts sampled during training. g) Evolution of constraint violation regret with respect to the target context distribution. h) Expected discounted cumulative cost of the final policies in contexts sampled during training. i) Expected success rate of the final policies in contexts sampled during training.

SCG-NoANN and SCG-NoPS do not fall behind in terms of pace of learning, we observe that SCG-NoPPPS is the slowest among all. In addition, SCG-NoPPPS yields high constraint violation regret. The last four subfigures visualize the progression of curricula generated by SCG variations.

## F.3    ABLATION STUDY FOR SCG PARAMETERS

We run an ablation study for three parameters of SCG: 1) target GMM weight ratio $\alpha$, i.e., the value SCG anneals $\alpha_k$ to, 2) number of annealing iterations $K_{ann}$, and 3) cost threshold $\tilde{D}$ in the SCG update. Our objective is to evaluate how SCG performs in safety-push, more specifically, how optimality at test time and safety during training change with respect to these parameters.

### F.3.1    TARGET GMM WEIGHT RATIO $\alpha$

Fig. 14 demonstrates the impact of target GMM weight ratio $\alpha$. We experiment with $\alpha \in \{0.75, 0.5, 0.875, 1\}$, which determines to what degree SCG prioritizes safe or performant contexts as the annealing progresses. $\alpha = 1$ corresponds to no annealing, i.e., SCG constructs source distributions $\varrho_+$ over safe contexts only in safety-prioritization phase (5) and over performant contexts only in performance-prioritization phase (6). In comparison, $\alpha = 0.5$ anneals the ratio to give equal importance to safety and performance. Hence, the safety-prioritization phase begins with safe contexts only and then anneals $\alpha_k$ to give equal importance. The performance-prioritization phase operates in reverse, first giving equal importance to both and then annealing it for performant contexts only. Note that $\alpha = 0.75$ is what we report in the main document. Although $\alpha \in \{0.875, 1\}$ provides lower CV regret than $\alpha = 0.75$ in multiple runs, $\alpha = 0.75$ achieves zero-cost and 100% success in target contexts, also yields CV regret with low variance. $\alpha = 0.5$ yields the highest CV regret.

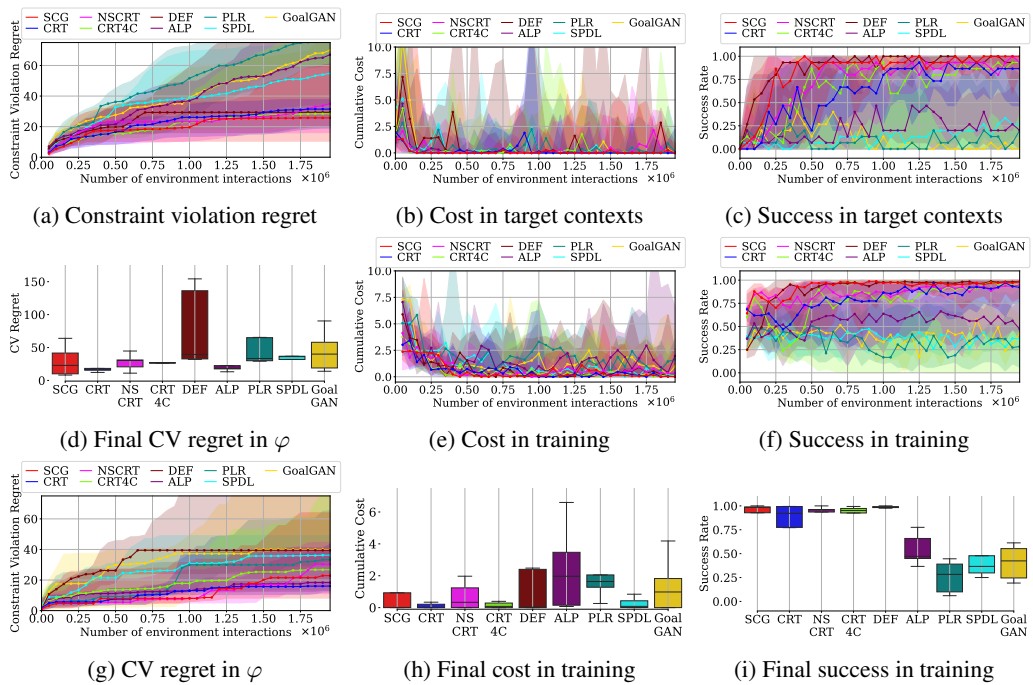

Figure 11: Safety-passage results from runs in 5 seeds: a) Evolution of constraint violation regret during. b) Progression of expected discounted cumulative cost in contexts drawn from the target context distribution. c) Progression of expected success rate in contexts drawn from the target context distribution. d) Constraint violation regret with respect to the target context distribution at the final curriculum iteration. e) Progression of expected discounted cumulative cost in contexts sampled during training. f) Progression of expected success rate in contexts sampled during training. g) Evolution of constraint violation regret with respect to the target context distribution. h) Expected discounted cumulative cost of the final policies in contexts sampled during training. i) Expected success rate of the final policies in contexts sampled during training.

### F.3.2 NUMBER OF ANNEALING ITERATIONS $K_{ann}$

Fig. 15 showcases the ablation results for the number of annealing iterations $K_{ann}$. This parameter specifies how many curriculum iterations the annealing of $\alpha_k$ to $\alpha$ takes. Note that we anneal $\alpha_k$ linearly. We experiment with $K_{ann} \in \{10, 5, 20\}$, where $\alpha = 10$ is reported in the main document. $K_{ann} = 10$ achieves zero-cost and 100% success rates in target contexts and provides low variance in CV regret. Decreasing or increasing the annealing pace causes a lower success rate and higher variance in CV regret.

### F.3.3 COST THRESHOLD $\tilde{D}$

Fig. 16 illustrates the ablation results for the cost threshold $\tilde{D}$. This parameter determines the maximum expected cost SCG allows a context under the new context distribution $\varrho_k$ (see equation 4). We experiment with $\tilde{D} \in \{1, 0.5, 2\}$, where $\alpha = 1$ is reported in the main document. $K_{ann} = 1$ achieves zero-cost and 100% success rates in target contexts and provides low variance in CV regret. Although $K_{ann} = 0.5$ can achieve lower CV regret, it significantly increases the variance and causes non-zero cost in target contexts in multiple runs.

### F.4 COMPARISON OF CONSTRAINED RL APPROACHES

In our experiments, all baselines, including DEFAULT, which does not generate curricula, and existing curriculum learning approaches use PPO-Lagrangian as the constrained RL algorithm of choice. To further support our claim that SCG provides safer training than DEFAULT, we experiment with three other constraint RL approaches in safety-push: 1) Constrained Policy Optimization (CPO),

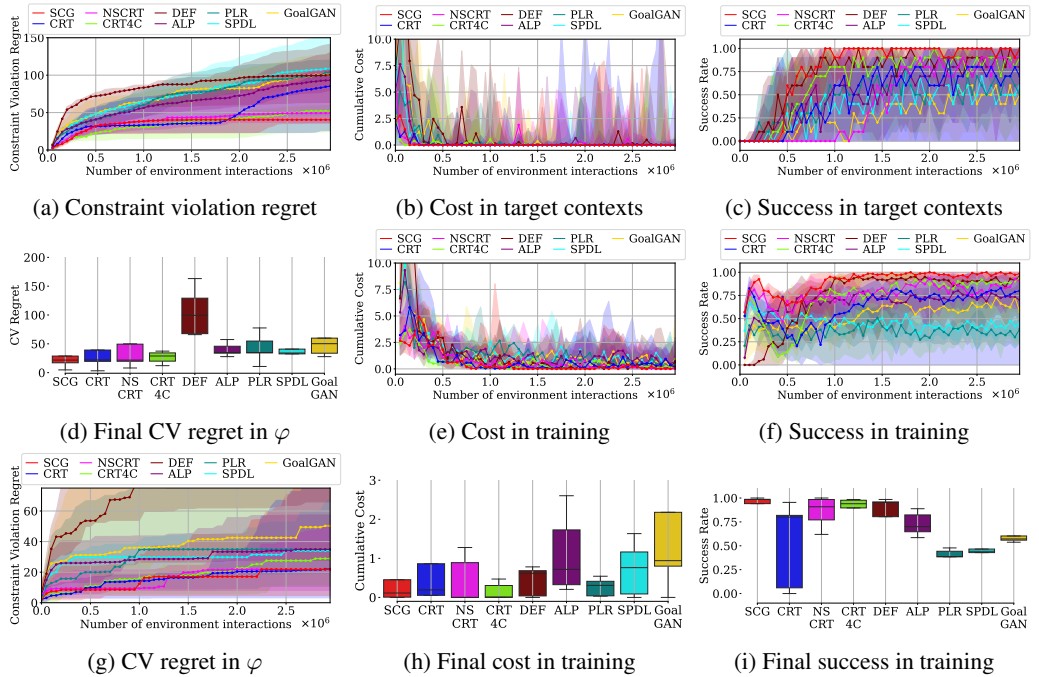

Figure 12: Safety-push results from runs in 5 seeds: a) Evolution of constraint violation regret during. b) Progression of expected discounted cumulative cost in contexts drawn from the target context distribution. c) Progression of expected success rate in contexts drawn from the target context distribution. d) Constraint violation regret with respect to the target context distribution at the final curriculum iteration. e) Progression of expected discounted cumulative cost in contexts sampled during training. f) Progression of expected success rate in contexts sampled during training. g) Evolution of constraint violation regret with respect to the target context distribution. h) Expected discounted cumulative cost of the final policies in contexts sampled during training. i) Expected success rate of the final policies in contexts sampled during training.

2) Projection-Based Constrained Policy Optimization (PCPO), and 3) First Order Constrained Optimization in Policy Space (FOCOPS). As DEFAULT, these approaches do not generate curricula but directly sample contexts from the target context distribution during training. Fig. 17 evidence that similar to DEFAULT with PPO-Lagrangian, other constrained RL algorithms cause high constraint violation regret during training and fail to yield optimal policies in multiple training runs.

## F.5 CURRICULUM PROGRESSION

Figs. 18 to 24 demonstrate the progression of curricula generated by CURROT, NAIVESAFECURROT, CURROT4COST, SPDL, PLR. ALP-GMM, and GOALGAN, respectively, in safety-maze, safety-goal, safety-passage, and safety-push environments. Figure 25 demonstrates the contexts drawn from the target context distribution during training runs of DEFAULT. NAIVESAFECURROT may have similar curricula to SCG in some settings, as it considers reward and cost simultaneously but through a penalized reward signal, which takes away the flexibility that SCG provides in prioritizing safe or performant contexts separately and sometimes together. CURROT4COST takes cost into account, only, but fails to recognize that goals on the hazards in safety-maze and safety-goal can lead to high constraint violation regret. ALP-GMM yields higher success than SPDL in safety-passage and safety-push, due to being able to generate multi-model distribution via Gaussian mixture models, as opposed to unimodal Gaussian distributions of SPDL. PLR fails to prioritize safe or performant contexts.

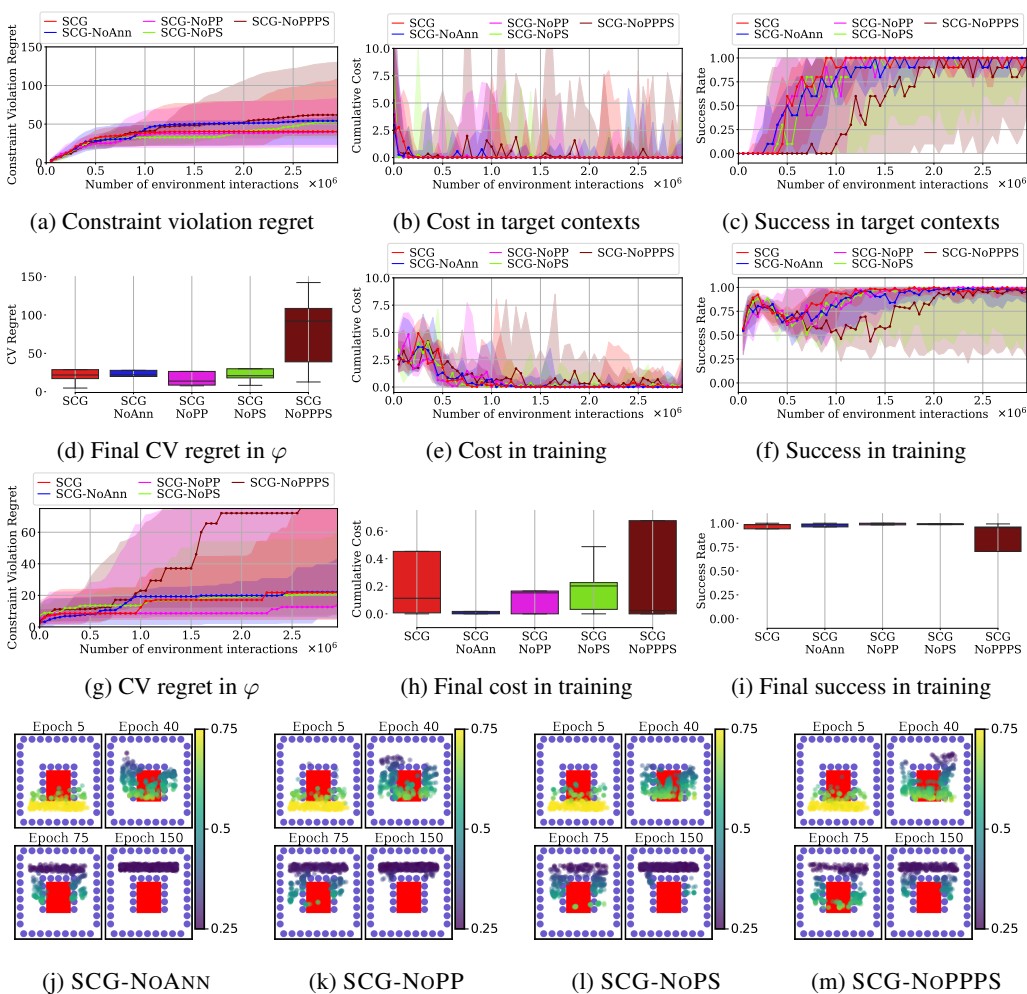

Figure 13: Ablation study results in safety-push from runs in 5 seeds: a) Evolution of constraint violation regret during. b) Progression of expected discounted cumulative cost in contexts drawn from the target context distribution. c) Progression of expected success rate in contexts drawn from the target context distribution. d) Constraint violation regret with respect to the target context distribution at the final curriculum iteration. e) Progression of expected discounted cumulative cost in contexts sampled during training. f) Progression of expected success rate in contexts sampled during training. g) Evolution of constraint violation regret in target contexts. h) Expected discounted cumulative cost of the final policies in contexts sampled during training. i) Expected success rate of the final policies in contexts sampled during training. j-m) Progression of curricula generated by SCG variations

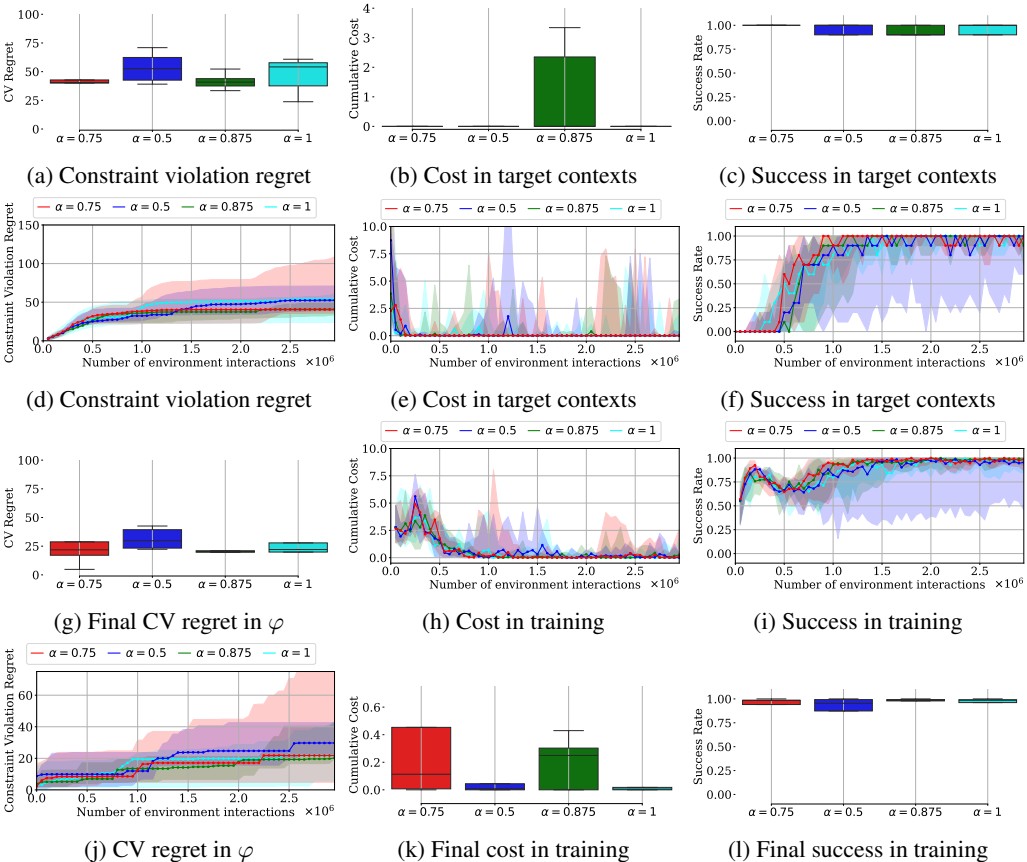

Figure 14: Ablation study for target GMM weight ratio $\alpha$ in safety-push from runs in 5 seeds: a) Constraint violation regret at the final curriculum iteration. b) Expected discounted cumulative cost of the final policies in target contexts, c) Expected success rate of the final policies in target contexts. d) Evolution of constraint violation regret during. e) Progression of expected discounted cumulative cost in contexts drawn from the target context distribution. f) Progression of expected success rate in contexts drawn from the target context distribution. g) Constraint violation regret with respect to the target context distribution at the final curriculum iteration. h) Progression of expected discounted cumulative cost in contexts sampled during training. i) Progression of expected success rate in contexts sampled during training. j) Evolution of constraint violation regret with respect to the target context distribution. k) Expected discounted cumulative cost of the final policies in contexts sampled during training. l) Expected success rate of the final policies in contexts sampled during training.

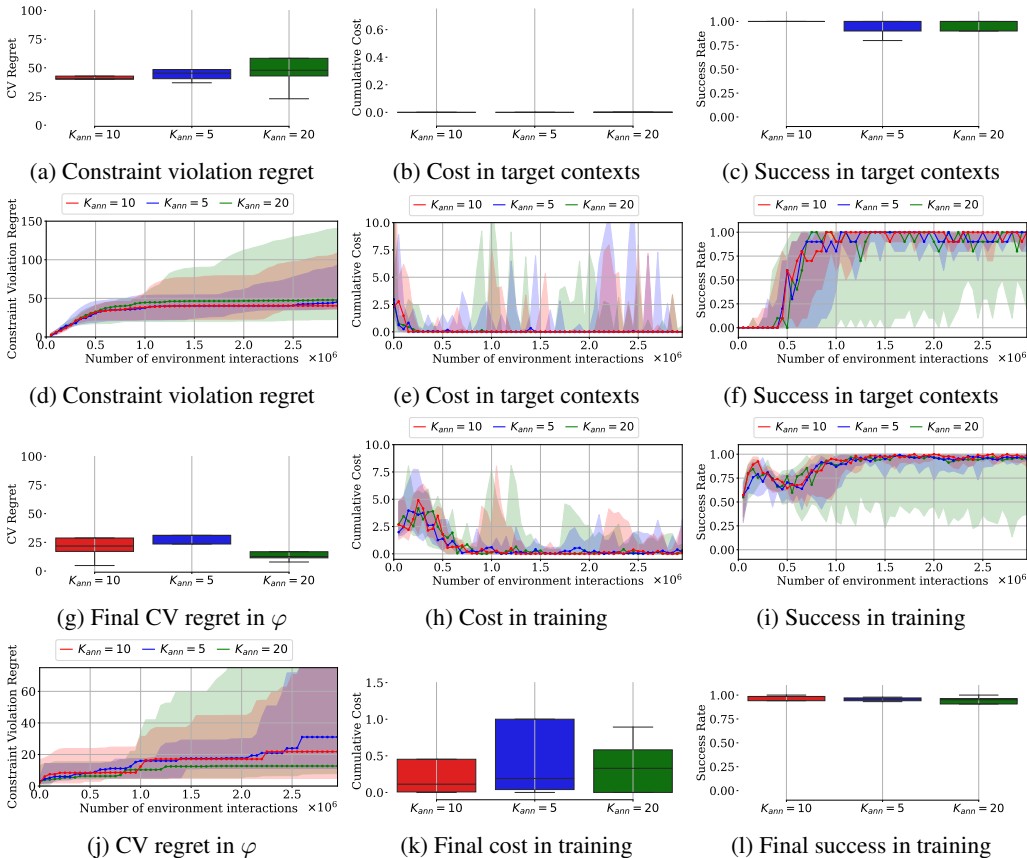

(a) Constraint violation regret  (b) Cost in target contexts  (c) Success in target contexts

(d) Constraint violation regret  (e) Cost in target contexts  (f) Success in target contexts

(g) Final CV regret in $\varphi$  (h) Cost in training  (i) Success in training

(j) CV regret in $\varphi$  (k) Final cost in training  (l) Final success in training

Figure 15: Ablation study for number of annealing iterations $K_{ann}$ in safety-push from runs in 5 seeds: a) Constraint violation regret at the final curriculum iteration. b) Expected discounted cumulative cost of the final policies in target contexts, c) Expected success rate of the final policies in target contexts. d) Evolution of constraint violation regret during. e) Progression of expected discounted cumulative cost in contexts drawn from the target context distribution. f) Progression of expected success rate in contexts drawn from the target context distribution. g) Constraint violation regret with respect to the target context distribution at the final curriculum iteration. h) Progression of expected discounted cumulative cost in contexts sampled during training. i) Progression of expected success rate in contexts sampled during training. j) Evolution of constraint violation regret with respect to the target context distribution. k) Expected discounted cumulative cost of the final policies in contexts sampled during training. l) Expected success rate of the final policies in contexts sampled during training.

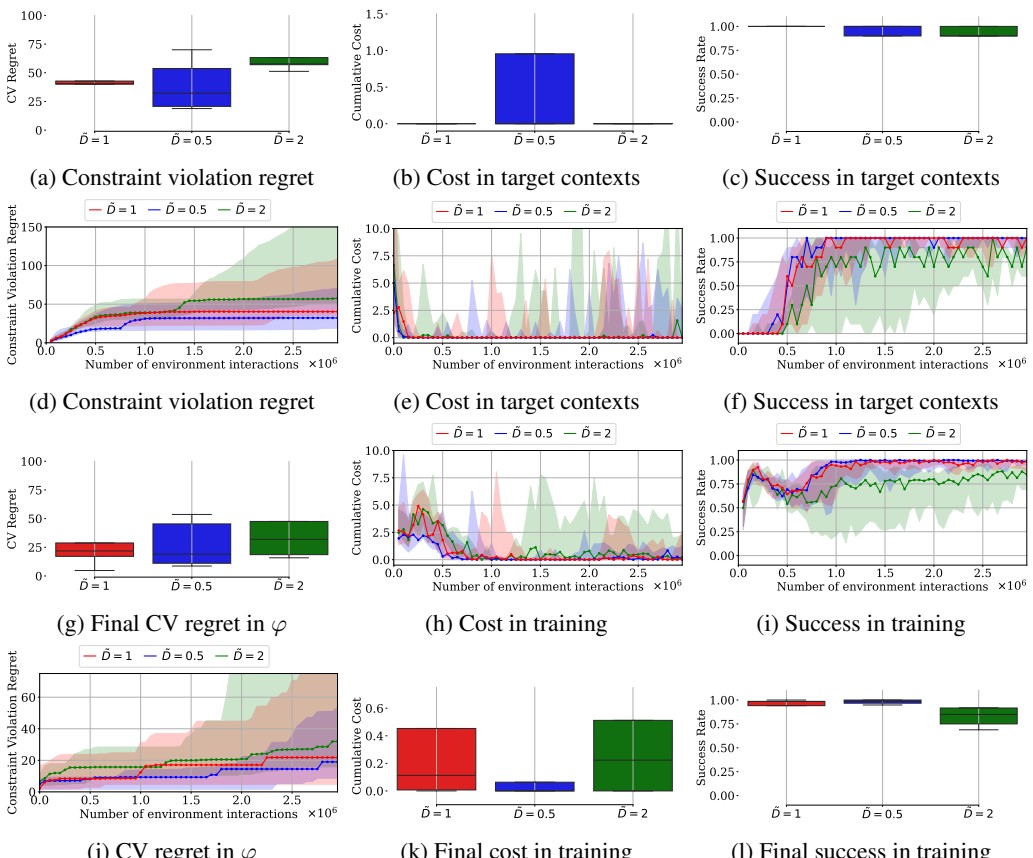

Figure 16: Ablation study for cost threshold $\tilde{D}$ in safety-push from runs in 5 seeds: a) Constraint violation regret at the final curriculum iteration. b) Expected discounted cumulative cost of the final policies in target contexts, c) Expected success rate of the final policies in target contexts. d) Evolution of constraint violation regret during. e) Progression of expected discounted cumulative cost in contexts drawn from the target context distribution. f) Progression of expected success rate in contexts drawn from the target context distribution. g) Constraint violation regret with respect to the target context distribution at the final curriculum iteration. h) Progression of expected discounted cumulative cost in contexts sampled during training. i) Progression of expected success rate in contexts sampled during training. j) Evolution of constraint violation regret with respect to the target context distribution. k) Expected discounted cumulative cost of the final policies in contexts sampled during training. l) Expected success rate of the final policies in contexts sampled during training.

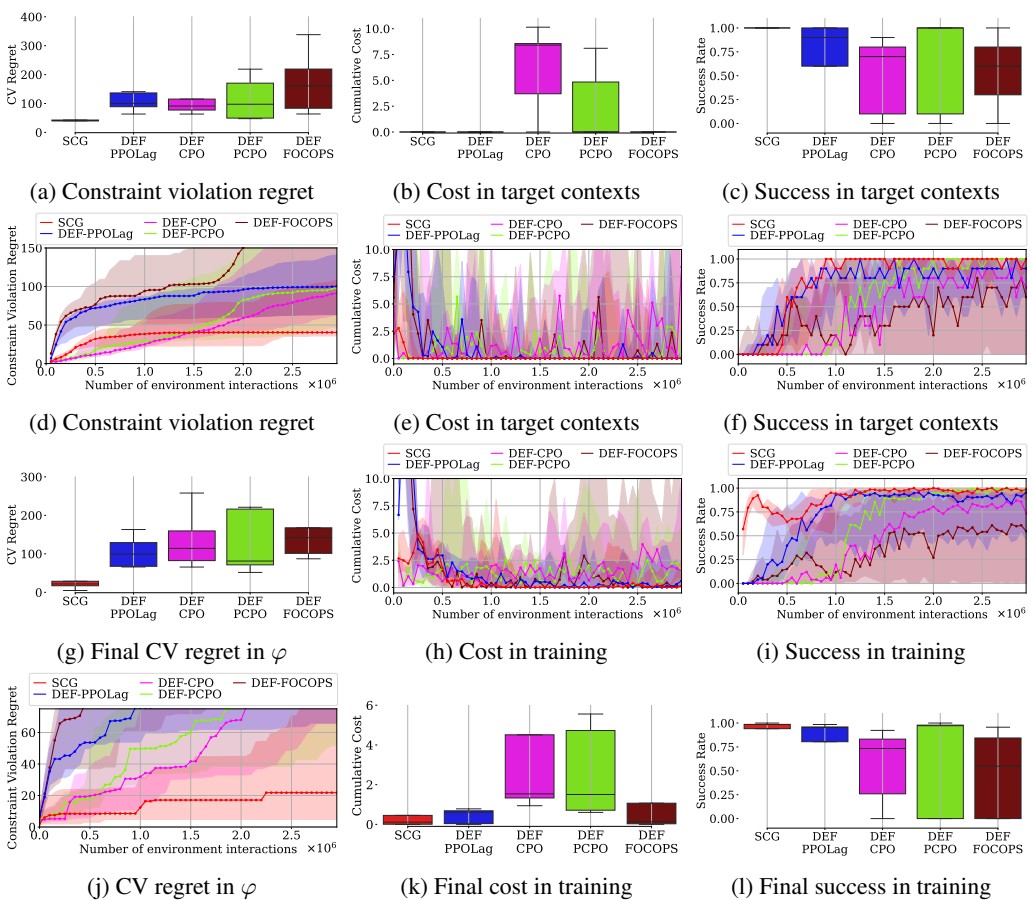

Figure 17: Comparison of constrained RL algorithms in safety-push from runs in 5 seeds: a) Constraint violation regret at the final curriculum iteration. b) Expected discounted cumulative cost of the final policies in target contexts, c) Expected success rate of the final policies in target contexts. d) Evolution of constraint violation regret during. e) Progression of expected discounted cumulative cost in contexts drawn from the target context distribution. f) Progression of expected success rate in contexts drawn from the target context distribution. g) Constraint violation regret with respect to the target context distribution at the final curriculum iteration. h) Progression of expected discounted cumulative cost in contexts sampled during training. i) Progression of expected success rate in contexts sampled during training. j) Evolution of constraint violation regret with respect to the target context distribution. k) Expected discounted cumulative cost of the final policies in contexts sampled during training. l) Expected success rate of the final policies in contexts sampled during training.

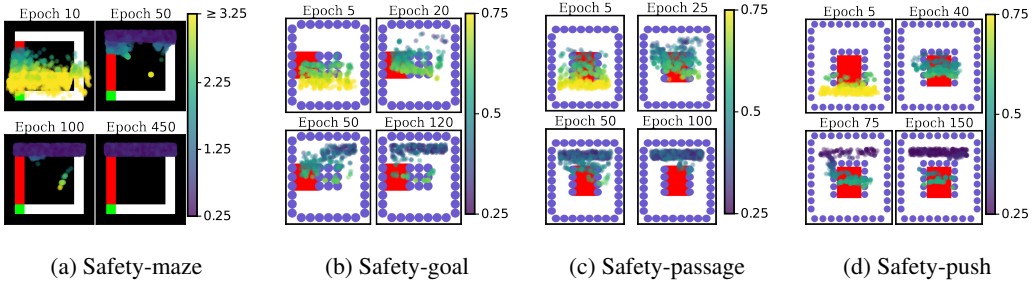

(a) Safety-maze     (b) Safety-goal     (c) Safety-passage     (d) Safety-push

Figure 18: Curricula generated by CURROT.

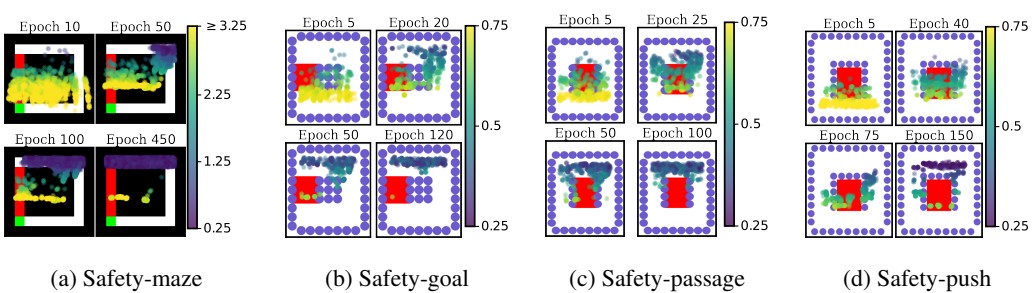

(a) Safety-maze     (b) Safety-goal     (c) Safety-passage     (d) Safety-push

Figure 19: Curricula generated by NAIVESAFECURROT.

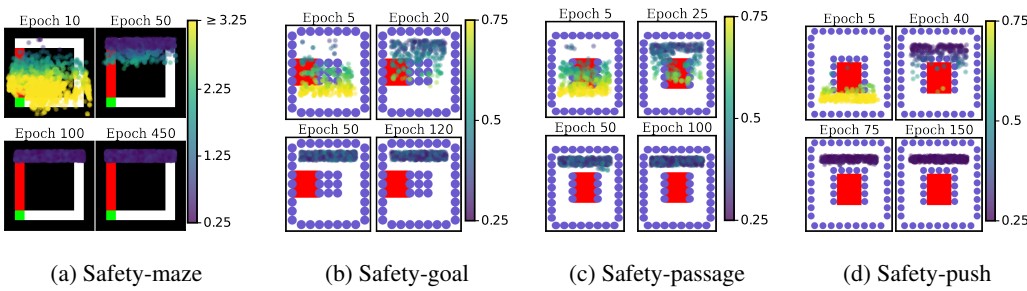

(a) Safety-maze     (b) Safety-goal     (c) Safety-passage     (d) Safety-push

Figure 20: Curricula generated by CURROT4COST.

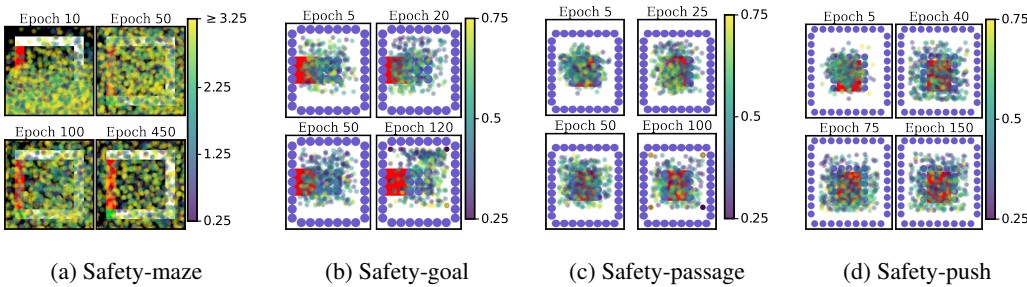

(a) Safety-maze     (b) Safety-goal     (c) Safety-passage     (d) Safety-push

Figure 21: Curricula generated by SPDL.

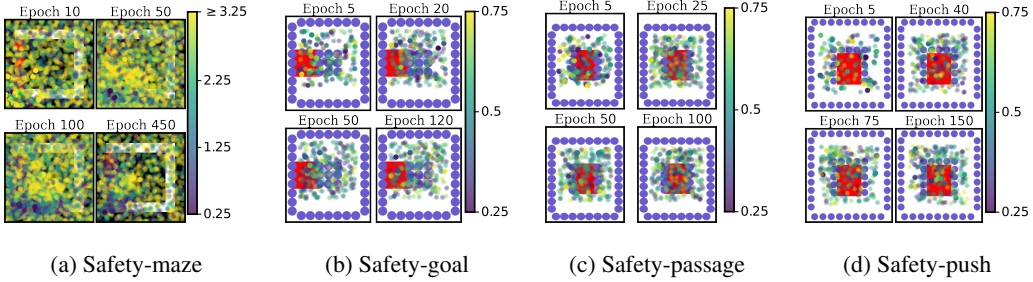

(a) Safety-maze      (b) Safety-goal      (c) Safety-passage      (d) Safety-push

Figure 22: Curricula generated by PLR.

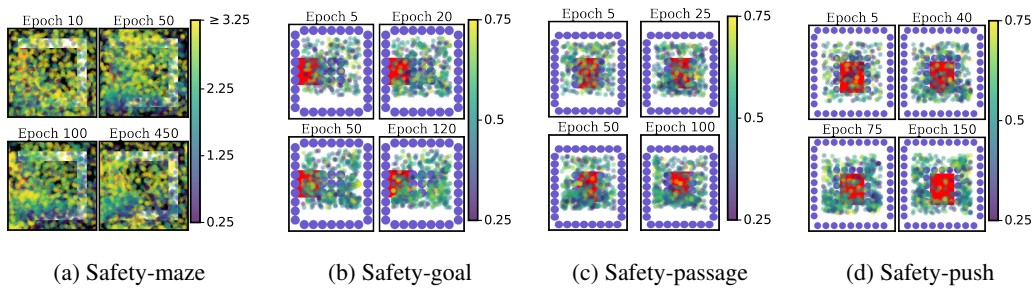

(a) Safety-maze      (b) Safety-goal      (c) Safety-passage      (d) Safety-push

Figure 23: Curricula generated by ALP-GMM.

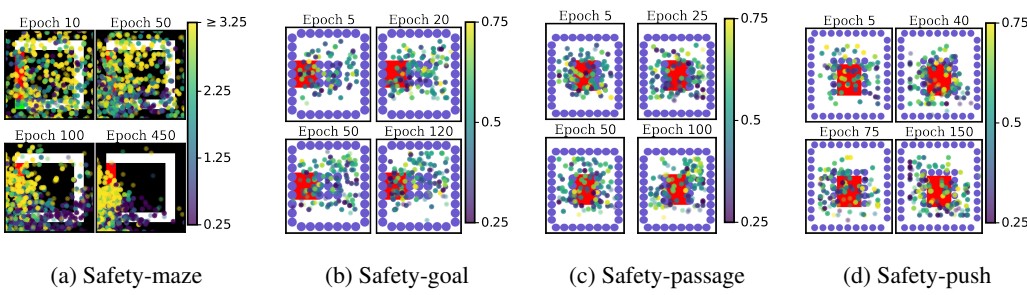

(a) Safety-maze      (b) Safety-goal      (c) Safety-passage      (d) Safety-push

Figure 24: Curricula generated by GOALGAN.

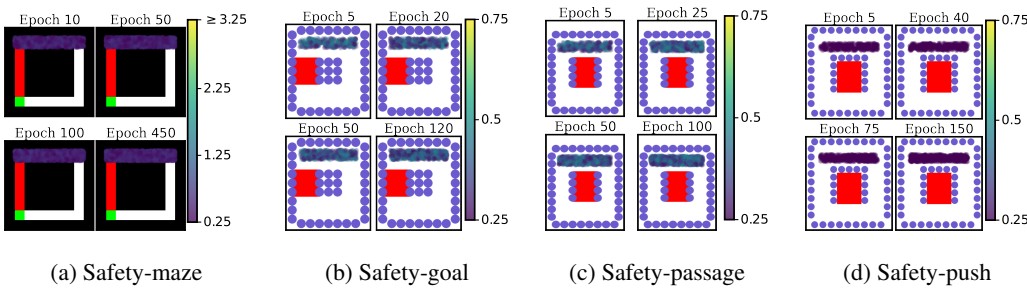

(a) Safety-maze      (b) Safety-goal      (c) Safety-passage      (d) Safety-push

Figure 25: Curricula generated by DEFAULT.

