# OpenReview forum: "Safety-Prioritizing Curricula for Constrained Reinforcement Learning"
_ICLR.cc/2025/Conference — ICLR 2025 Poster_

### Official Review · Reviewer_SUye · 2024-10-31

**Soundness:** 2
**Presentation:** 2
**Contribution:** 2
**Rating:** 6
**Confidence:** 3

**Summary:**

This paper introduces a safe curriculum generation (SCG) approach. Using safety constraints, SCG constraints the evolution of the curriculum to stay in the safe zone while annealing towards the maximization of the policy's performance. Based on the CURROT, SCG includes safety constraints to ensure that agent will not venture and make dangerous movements during training. Focusing on Safety-Push, Safety-Maze, and Safety-Goal environments, the authors explores how such design choices makes SCG outperform CURROT and other baselines.

**Strengths:**

This paper is generally well written and the structures are easy to follow, with the headers clearing marking out the points to be made across. The relative work is quite comprehensive, and extensive experiments conveys the main points well.

**Weaknesses:**

- The algorithm is heavily inspired by CURROT and it feels like a somewhat of a safe RL implementation of CURROT. While some may consider this to be a incremental improvement of algorithm, I think the paper could be made stronger if the authors could share the design insights that led to SCG. For example, why did the authors decided to design an algorithm inspired by CURROT? The algorithmic features that differentiates SCG from CURROT, what were their motivations? If the authors could summarize their insight in a way that could help other researchers to make improvements to other existing algorithms in RL, the contribution of this paper would be much more significant.

- I think the paper does not convey well some of the underlying assumptions and motivations, especially towards the empirical analysis. I have included the details in the questions.

**Questions:**

* What is the motivation for choosing the safety-maze, safety-goal, and safety-push as benchmarks? Why would you consider that these baselines would demonstrate the pros and cons of SCG better than other benchmarks?
* Is the 'Default' mentioned in the empirical studies results of running RL without any curriculum generator?
* As this algorithm is inspired by CURROT, I understand why CURROT was chosen as baselines. But as for the rest, why were the other baselines selected for? What does comparing SCG with the selected baselines demonstrate?
* In figure 4, we see boxes and bars. Would it be correct to assume that boxes are standard deviations and bars are the maximum and minimum of the data distribution?

---

> ### Author Response · Authors · 2024-11-20
> **Response to Reviewer SUye**
>
> We would like to thank the reviewer for their thought-provoking comments and questions, which helped us improve our work in the rebuttal phase. Below, we respond to their comments regarding the weaknesses and then their questions.
>
> **W1:** Lack of reasoning for design insights
>
> We agree with the reviewer that our work could help other researchers by clarifying why we develop SCG on top of CURROT. To this aim, we added the sentences below to Section 4.1:
>
> > We investigate CURROT due to its favorable properties 1) It poses curriculum generation as a constrained optimization problem, allowing for a natural extension to CCRL (1) 2) Interpolating context distributions based on Wasserstein distance enables non-parametric distributions. 3) The performance constraint enforces a desired return level in any context, hence robustness.
>
> Three novel algorithmic features differentiate SCG from CURROT:
> * Cost constraint in the curriculum update in Eq. (4) directly accounts for safety in any context.
> * Safety-prioritization via generating source distributions over safe contexts in Eq. (5) enables focusing on low-cost contexts early during training to avoid high constraint violation regret.
> * Annealing mechanism to tune the prioritization ratio of safe or performant contexts in Eq. (5) and (6) allows for a smooth transition between safety and performance prioritization phases.
>
> We also made changes in Section 5 to highlight the motivation behind these algorithmic differences.
>
> **W2:** Lack of explanation of assumptions and motivations
>
> We address this section in the corresponding questions.
>
> **Q1**: What are the motivations behind the environments of interest?
>
> We study these environments for the following reasons:
> * **Highlighting the misalignment problem between curriculum learning and constrained RL:** In all domains, there is a trade-off between unsafe yet short (high return and high cost) paths over hazards and safe yet long (low return and low cost) paths over clear zones.
> * **Safety-Maze is a cost-constrained version of Maze**, a domain studied in the paper that developed CURROT. We choose Maze because it is a simple domain, yet the cost modification to present the misalignment problem makes CURROT an unsafe curriculum learning approach.
> * **Safety-Goal and Safety-Maze are navigation tasks with realistic sensory observations** in Safety-Gymnasium, a framework extensively used for constrained RL.
>
> The second paragraph in Section 6 explains the challenges introduced in Safety Push, such as the need for multi-modal context distributions or the dead-end causing high constraint violation regret.
>
> In addition, these environments differ from each other in terms of the following aspects:
> * **Dimensionality of state space**: 2D (coordinates) in Safety-Maze, whereas 72D and 88D (LIDAR sensor outputs) in Safety-Goal and Safety-Push,
> * **Complexity of dynamics**: Safety-Maze has easy dynamics, yet the cars in Safety-Goal and Safety-Push need to maneuver around hazards and pillars,
> * **Types of reward functions**: Safety-Maze has a sparse reward function (1 for reaching the goal and zero for all other transitions), whereas Safety-Goal and Safety-Push utilize the displacement-based reward functions in Safety-Gymnasium.
> * **Objects to interact with**: In Safety-Push, the car must learn to attach its front end to the box at the right angle and then push the box around pillars to the goal.
>
> **Q2:** Does DEFAULT not generate a curriculum?
>
> Yes, DEFAULT directly draws context from the target context distribution and runs a constrained RL algorithm, specifically PPO-Lagrangian, without generating a curriculum. We made a minor clarification in Section 6 to address this question.

---

> ### Author Response · Authors · 2024-11-20
> **Response to Reviewer SUye**
>
> **Q3:** What is the reasoning behind baselines other than CURROT?
>
> DEFAULT allows us to understand whether a curriculum learning approach has benefits in terms of safety during training or optimality at test time.
>
> CURROT4Cost replaces CURROT's performance constraint with a cost constraint. Therefore, we evaluate whether the advantages of SCG can be obtained via a curriculum learning approach that only cares about safety.
>
> NaiveSafeCURROT penalizes the reward with cost instead of return in the performance constraint of CURROT. NaiveSafeCURROT enables us to investigate whether a naive attempt at modifying CURROT to be safe is sufficient.
>
> SPDL is also an interpolation-based curriculum learning approach that poses curriculum generation as a constrained optimization problem. Compared to CURROT, SPDL minimizes the KL divergence from the target context distribution, and its performance constraint is based on the expected return with respect to the subsequent context distribution. SPDL generates parametric context distributions and does not provide the robustness CURROT achieves via a performance constraint on individual constraints. Therefore, we can highlight why CURROT is a superior algorithm for building SCG.
>
> PLR is a popular curriculum learning approach, yet it is unsuitable for continuous context spaces. It assumes that contexts are discrete and unidimensional and that the target distribution is a uniform context distribution.
>
> ALP-GMM and GoalGAN are among the first automated curriculum generation approaches. They generate a curriculum as a sequence of distributions, similar to CURROT and SPDL, yet they do not interpolate them via Wasserstein distance or KL divergence. Similar to PLR, they assume uniform target context distributions.
>
> **Q4:** What do the box plots indicate?
>
> We thank the reviewer for their attention to detail. We accidentally commented on this sentence in the first submission. We added the following sentence to the caption of Figure 4:
>
> > Box plots show the minimum, the first quartile, the median, the third quartile, and the maximum, from bottom to top.

---

> > ### Comment · Reviewer_SUye · 2024-11-25
> >
> > Thank you for your response! I feel that my questions has been adequately responded and I have changed my scores accordingly.

---

> > > ### Author Response · Authors · 2024-11-25
> > > **Thank you!**
> > >
> > > Thank you for reading our rebuttal and raising your score towards acceptance!

---

### Official Review · Reviewer_izzQ · 2024-11-03

**Soundness:** 3
**Presentation:** 3
**Contribution:** 2
**Rating:** 6
**Confidence:** 5

**Summary:**

This paper presents a safe curriculum generation (SCG) approach aimed at enhancing safety during training while improving sample efficiency. SCG generates task sequences as context distributions, prioritizing tasks with minimal safety violations over high-reward tasks in the early stages to safely converge on the target context distribution. Empirical studies indicate that SCG outperforms state-of-the-art curriculum learning methods and their naively adapted safe versions, achieving optimal performance with the fewest constraint violations during training.

**Strengths:**

- The paper effectively highlights the gap in current curriculum learning (CL) algorithms and the importance of developing safe CL approaches.
- Background and related work are thoroughly described, making the contribution and method easy to understand.
- The paper is well-organized, with clear explanations and logical flow.
- The algorithm demonstrates state-of-the-art performance across multiple benchmarks.

**Weaknesses:**

- Limited Novelty: The approach’s novelty is modest, as it primarily adds an additional constraint to the existing CURROT algorithm.
- Empirical Limitations:
    - The empirical evaluation lacks comparisons with other safe-RL baselines and is limited to three domains.
    - Sensitivity to newly introduced parameters (e.g., \apha_k) is not thoroughly analyzed.

**Questions:**

- Was the cost annealed in the Naive approach as well?
- How does the proposed approach compare to other safe-RL algorithms in these domains?

---

> ### Author Response · Authors · 2024-11-20
> **Response to Reviewer izzQ**
>
> We would like to thank the reviewer for their fair criticism and helpful questions that allowed us to improve the presentation of our work.
>
> Below, we address the reviewer's comments regarding weaknesses first, then their questions:
>
> **W1:** Limited novelty.
>
> SCG not only adds a cost constraint to CURROT but also has two other features to achieve safer training and optimality at test time, as our ablation study in Section 6.3 indicates.
>
> SCG's three novel features built on top of CURROT:
> * Cost constraint in the curriculum update in Eq. (4).
> * Safety-prioritization via generating source distributions over safe contexts in Eq. (5).
> * Annealing mechanism to tune prioritization ratio of safe or performant contexts in Eq. (5) and (6).
>
> Our ablation results in Figure 6 showcase that all these features result in a well-balanced, safe curriculum learning approach. Simply adding the cost constraint without safety-prioritization (SCG-NoPS and SCG-NoPPPS) may make the training safer than CURROT (see the bottom row of Figure 4a) but does not achieve optimality, as SCG-NoPS fails to yield zero cost in multiple runs and SCG-NoPPPS cannot achieve 100% success. Furthermore, not having annealing (SCG-NoAnn) causes higher CV regret and a lower success rate than SCG with all three features (see Figures 6a and 6c).
>
> **W2:** Empirical limitations.
>
> As indicated by the last sentence before Section 6.1, our empirical analysis involves a comparison against constrained RL algorithms that do not use curriculum learning. Appendix F.3. demonstrates an empirical comparison of SCG against PPO-Lagrangian, Constrained Policy Optimization (CPO), Projection-based CPO (PCPO), and First-Order Constrained Optimization in Policy Space (FOCOPS) in Safety-Push. Figure 14 evidences that, similar to DEFAULT, which uses PPO-Lagrangian without curriculum generation, CPO, PCPO, and FOCOPS cause high constraint violation regret during training and fail to yield optimal policies in multiple training runs.
>
>
> Following the suggestion of the reviewer about sensitivity analysis to parameters of SCG, our general response includes an ablation study on three parameters:
> * Target Gaussian mixture model weight ratio $\alpha$, i.e, the target for annealing $\alpha_k$,
> * Number of annealing steps $K_{ann}$,
> * Cost threshold $\tilde{D}$ in SCG update
>
> **Q1:** Cost annealing in the Naive approach
>
> No, there is no annealing in NaiveSafeCURROT because NaiveSafeCURROT does not have separate safety and performance prioritization phases. NaiveSafeCURROT extends CURROT by penalizing rewards with costs.
>
> **Q2:** How does the proposed approach compare to other safe-RL algorithms in these domains?
>
> As our response to W2 indicates, Appendix F.3 compares SCG against constrained RL algorithms.
> In summary, without curriculum generation, CPO, PCPO and FOCOPS cause high constraint violation regret during training and fail to yield optimal policies in multiple training runs.

---

> > ### Comment · Reviewer_izzQ · 2024-11-25
> >
> > I thank the authors for their detailed rebuttal and their efforts to address my concerns. While I appreciate the additional clarifications and results provided, I still find the novelty of the proposed approach to be somewhat limited, as the modifications to CURROT, such as adding a safety constraint and annealing, appear relatively incremental. Furthermore, the evaluation could be improved by including comparisons with curriculums optimized using safe RL algorithms and testing on popular benchmark domains. These additions would better demonstrate the approach’s broader applicability and effectiveness. For these reasons, I will not be changing my score.

---

> > > ### Author Response · Authors · 2024-11-28
> > >
> > > We thank the reviewer for their response and explaining the reasoning behind their decisions.
> > >
> > > We want to make a correction. **In our evaluations, we combine all curriculum learning approaches with a safe RL algorithm**, as lines 364-365 indicate. In other words, a curriculum learning algorithm generates a sequence of tasks, and the RL agent optimizes its policy by interacting with these tasks via a safe RL algorithm. Our experiments use PPO-Lagrangian as the safe RL algorithm of choice.
> > >
> > > Our work is the first to illuminate the misalignment between curriculum learning and constrained RL (see Section 4.2). Because no benchmark domains exist in the curriculum learning literature that contributes to this phenomenon, we designed Safety-Maze, Safety-Goal, and Safety-Push.
> > >
> > > We study Safety-Maze to showcase that a simple modification to an existing domain in curriculum learning, i.e., Maze [1], can cause the misalignment phenomenon. We design Safety-Goal and Safety-Push in Safety-Gymnasium [2], a popular benchmark for constrained RL, to have realistic observations and complex dynamics and tasks. Similar to what we observe in Safety-Maze, all other evaluated approaches fail to yield optimal policies or provide safety during training due to the misalignment phenomenon. We note that Safety-Gymnasium does not have contextualized domains by default.
> > >
> > > - **References**
> > >     1. Klink, P., Yang, H., D’Eramo, C., Peters, J., & Pajarinen, J. (2022, June). Curriculum reinforcement learning via constrained optimal transport. In International Conference on Machine Learning (pp. 11341-11358). PMLR.
> > >     2. Ray, A., Achiam, J., and Amodei, D. (2019). Benchmarking safe exploration in deep reinforcement learning.

---

> > > > ### Comment · Reviewer_izzQ · 2024-11-28
> > > >
> > > > I thank the authors for their detailed clarification regarding the use of safe RL algorithms in conjunction with curriculum learning approaches, as well as the explanation of the custom benchmark domains. This has addressed some of my earlier concerns. Based on this clarification, I will be increasing my score.

---

> > > > > ### Author Response · Authors · 2024-11-28
> > > > >
> > > > > We thank the reviewer for their response and for increasing their score towards acceptance!

---

> ### Author Response · Authors · 2024-11-25
> **A kind reminder**
>
> In our rebuttal, we addressed the reviewer's comments on
> - novelty by describing the algorithmic contributions of SCG and
> - empirical limitations via a new ablation study and clarifying that we have already compared SCG with other safe-RL baselines.
>
> We hope we addressed all their comments/questions sufficiently well and would appreciate it if the reviewer could point out any further comments/questions. Otherwise, we kindly ask them to consider revising their score accordingly.

---

### Official Review · Reviewer_RChd · 2024-11-04

**Soundness:** 3
**Presentation:** 4
**Contribution:** 2
**Rating:** 6
**Confidence:** 4

**Summary:**

This paper proposes modifying curriculum generation techniques to incorporate safety constraints. It does so by constraining the curriculum to not produce tasks which tend to result in constraint violations.

**Strengths:**

The clear exposition quickly brings the reader up to speed with the pragmatic approaches in the space of curriculum generation. It makes it clear how safety constraints are not incorporated in current techniques, providing a clear gap for this method to fill. The method appears to work approximately as advertised, resulting in similar performance to prior works with less violation of constraints.

**Weaknesses:**

The biggest weakness of this work is the lack of a demonstration of scalability. The 2-d navigation task with 2-d goals provides a very small space for curriculum design where all tasks can be approximately enumerated. It is unclear how this approach would scale to situations where the spaces of possible tasks is combinatorially large. However, as this is a concern that is shared with much of the prior work, I do not see this as a reason in itself to reject the paper.

The main limitation keeping me from raising my score is that it appears the impact of this work is currently somewhat limited. This is in-part because of the scale, in part because the gains on both safety and performance are not as dramatic as one would hope, and in part because it appears to be far from the point where it could be tested in concrete applications. My evaluation of this paper would be significantly higher if it were shown to be applicable to a robotics domain, where safety during training is a critical component. I am guessing this is not in the current draft because it is difficult to scale underlying methods like CURROT to those domains, but if that is the case, then it is not clear if this approach would still be applicable with whatever approach does eventually scale. These uncertainties make it less clear for the reader what to take away from this paper for their own future research.

The one point where the paper can be made significantly more clear is explaining the apparently constraint-violating tasks in Figure 5 c and Figure 3 (right). My sense is that these tasks are not as constraint-violating as they seem since they have a high-tolerance, but it cannot be confirmed from the diagram since there is no correspondence between color and radius of the tolerance. I think this could be resolved by adding a few circles to-scale to ones side of these figures which visually depicts what color would correspond to a given radius. This could then let me check that the tolerance of the radius is sufficient to let the agent avoid a constraint violation.

Minor point:
- 188 and 193 I think these sentences do not completely describe the objective, since they does not say what happens when "finding the optimal policy" and "minimising the constrained violation regret" are in tension.

**Questions:**

What is the main limitation to applying this approach to higher-dimensional environments like Brax?

---

> ### Author Response · Authors · 2024-11-20
> **Response to Reviewer RChd**
>
> We would like to thank the reviewer for their positive comments, valuable criticism, and thought-provoking questions that helped us improve the presentation of our work.
>
> We first respond to the reviewer's comments and then their question.
>
> **W1:** Lack of a demonstration of scalability
>
> Indeed, we study low-dimensional context spaces, specifically three-dimensional contexts, that determine the position and tolerance of goals. As the reviewer indicates, this is a common point of interest in the existing automated curriculum generation work, such as CURROT[1] and GRADIENT[2]. The main reason behind this is that off-the-shelf RL algorithms can fail even in multi-task settings where contexts are low-dimensional. An interesting open question in curriculum learning is how to address a high-dimensional context space, e.g., a space comprised of goal images [2].
>
> We consider the scalability of curriculum generation to higher-dimensional context spaces orthogonal to the safety curriculum generation. We believe that techniques such as SCG will benefit from novel curriculum generation methods that scale to larger context spaces.
>
> We also want to note that Safety-Goal and Safety-Push are not two-dimensional navigation tasks. The agent observes 72D and 88D observations comprised of LIDAR outputs for surrounding objects.
>
> **W2:** Gains on both safety and performance are not as dramatic.
>
> Could the reviewer clarify what they mean by 'dramatic' if possible?
>
> Table 1 in Section 6.1. highlights that SCG is the only approach that yields zero cost in target contexts (Fig. 4b), hence satisfies the cost constraint in the CCRL objective while achieving the highest success rate in all domains (Fig. 4c). Simultaneously, SCG achieves the lowest constraint violation regret (Fig. 4a) among approaches that achieve zero cost and highest success. These results evidence that failure to address the misalignment phenomenon, regardless of the domain complexity, causes suboptimal policies or optimal policies at the expense of high constraint violation regret.
>
> **W2:** Applicability to robotics domains
>
> We agree that studying domains consisting of complex rigid bodies in Brax would further strengthen our empirical analysis. However, we would like to highlight that the studied domains in this paper are not necessarily simple and unrealistic. They differ in terms of the following aspects:
> * **Dimensionality of observation spaces**: 2D (coordinates) in Safety-Maze, whereas 72D and 88D (LIDAR sensor outputs) in Safety-Goal and Safety-Push,
> * **Complexity of dynamics**: Safety-Maze has easy dynamics, yet the cars in Safety-Goal and Safety-Push need to maneuver around hazards and pillars,
> * **Types of reward functions**: Safety-Maze has a sparse reward function (1 for reaching the goal and zero for all other transitions), whereas Safety-Goal and Safety-Push utilize the displacement-based reward functions in Safety-Gymnasium.
> * **Objects to interact with**: In Safety-Push, the car must learn to attach its front end to the box at the right angle and then push the box around pillars to the goal.
>
> We study Safety-Maze to showcase that a simple modification to an existing domain in curriculum learning can cause the misalignment phenomenon. As the domain complexity increases in Safety-Goal and Safety-Push, existing baselines and naive attempts to make them safe fail to yield optimal policies and provide safety during training. Safety-Goal and Safety-Push are navigation tasks with realistic sensory observations in Safety-Gymnasium [3], a framework extensively used for constrained RL.
>
> Although some rigid body domains in Brax have higher dimensional observation and action spaces, this does not necessarily mean that CURROT-like approaches cannot scale to them, as observation/action spaces differ from context spaces. In case one is interested in a high dimensional context space, where, for example, goals are indicated by positions/angles of tens of joints, then this would indeed increase the computational burden of the curriculum update in Eq. (4). Nevertheless, CURROT-like approaches can still address such settings. In Appendix A, the CURROT paper [1] describes how the library they utilize in their implementation can solve the optimal transport problem for sets of hundreds of thousands of samples in seconds on high-end GPUs [4]. More specifically, approximate solutions lower the complexity $\mathcal{O}(n^3)$ to $\mathcal{O}(Mn\log n)$ where $n$ is the dimensionality of a context space and $M\ll n$ [5,6].
>
> **W4:** Particle colors in Figures 3 and 5c (right) are hard to interpret.
>
> In our global response, we have provided new curriculum progression plots that use varying dimensions instead of varying colors for goal tolerances, as requested by the reviewer.

---

> ### Author Response · Authors · 2024-11-20
> **Response to Reviewer RChd - 2**
>
> **W5:** Lines 188 and 193 do not completely describe the objective.
>
> We want to thank the reviewer for this helpful suggestion. We made a minor change in the problem statement section to mention the misalignment phenomenon before going into detail in Section 4.2.
>
> **Q1:** What is the main limitation to applying this approach to Brax environments?
>
> Our response to the reviewer's comments explains why applying CURROT-like algorithms to Brax environments may not be a major limitation. We also describe the reasoning behind choosing our environments of interest.
>
> Nevertheless, we want to acknowledge that evaluation in complex robotics domains is a common missing piece in curriculum learning literature. We are working on addressing this evaluation gap in our current projects.
>
> - **References**
>     1. Klink, P., Yang, H., D’Eramo, C., Peters, J., & Pajarinen, J. (2022, June). Curriculum reinforcement learning via constrained optimal transport. In International Conference on Machine Learning (pp. 11341-11358). PMLR.
>     2. Huang, P., Xu, M., Zhu, J., Shi, L., Fang, F., & Zhao, D. (2022). Curriculum reinforcement learning using optimal transport via gradual domain adaptation. Advances in Neural Information Processing Systems, 35, 10656-10670.
>     3. Ji, J., Zhang, B., Zhou, J., Pan, X., Huang, W., Sun, R., ... & Yang, Y. (2023). Safety gymnasium: A unified safe reinforcement learning benchmark. Advances in Neural Information Processing Systems, 36.
>     4. Feydy, J. and Roussillon, P. Geomloss, 2019. URL https://www.kernel-operations.io/ geomloss/index.html.
>     5. Bonneel, N., Rabin, J., Peyré, G., & Pfister, H. (2015). Sliced and radon wasserstein barycenters of measures. Journal of Mathematical Imaging and Vision, 51, 22-45.
>     6. Kolouri, S., Nadjahi, K., Simsekli, U., Badeau, R., & Rohde, G. (2019). Generalized sliced wasserstein distances. Advances in neural information processing systems, 32.

---

> ### Author Response · Authors · 2024-11-28
> **A kind reminder**
>
> We thank the reviewer for initiating a discussion on the scalability of our proposed approach and its applicability to robotics domains, as well as for helping us improve the presentation of our work. As the end of the discussion period approaches, we would like to ask the reviewer if they have any remaining questions.

---

> ### Comment · Reviewer_RChd · 2024-11-28
> **Response to Rebuttal**
>
> I thank the authors for their thoughtful rebuttal, and clarifying figures. I agree with most of the response and do not have any remaining questions. Though I believe this paper to be correct, I find it difficult to recommend it more strongly without an empirical demonstration of scalability. Thus I will, unfortunately, be leaving my score unchanged.

---

### Official Review · Reviewer_Neqb · 2024-11-04

**Soundness:** 3
**Presentation:** 3
**Contribution:** 2
**Rating:** 3
**Confidence:** 3

**Summary:**

The paper tried to ensure the safety of curriculum learning.
Unlike previous curriculum learnings that tried to make the agent perform better, this paper prioritized safety but still cared about performance.
The paper compared their results with CURROT(Klink 2022) on safety maze, safety-goal, and safety-push, which did not care about safety.
The results showed that SCG can achieve a low CV regret and still achieve a high success rate.

**Strengths:**

The performance seems to be better.
It can achieve optimal policy.
Compared to CURROT,  it is safer.
Compared to DEFAULT, it is more sample efficiency.

**Weaknesses:**

1 The paper only compared simple environments. It does not compare to the environments in the previous work. For example, Point Mass and Bipedal Walker Stump Tracks.

2 The method seems trivial to me. It is mostly based on CURROT.

3 There are many other curricular learning methods with different settings. This paper can show that it can be used in the same setting as CURROT.

**Questions:**

1 A proper baseline should be using a goal that cares about both safety and performance. The score can weighted sum of both scores. Using different weights will be interesting.

2 Since the paper cares about cumulated regret, have you thought about the possibility of making more mistakes first and learn those mistakes can avoid future mistakes?

---

> ### Author Response · Authors · 2024-11-20
> **Response to Reviewer Neqb**
>
> We thank the reviewer for taking the time to write a critique of our work. Below, we address their comments and questions.
>
> **W1:** 'Simple' environments and lack of comparison in previously studied domains
>
> We would like to point out that the environments were carefully chosen to be challenging for constrained RL agents.
> Notice, for instance, that the Default approach, which does not use a curriculum, often fails to solve the underlying task (see Fig 4.c in the paper).
>
> Furthermore, the tasks considered have a clear misalignment between the reward and costs. They force the agents to find a tradeoff between the reward and cost to satisfy the constraints, such that simply maximizing the reward does not return a policy that satisfies the constraints.
> From this perspective, the Bipedal Walker Stump is not interesting for our work, as the cost is already associated with falling over, and to maximize the return, the agent should avoid falling over anyway.
>
> Furthermore, we would like to show that the environments studied are not 'simple', as they present varying challenges:
> * **Dimensionality of state space**: 2D (coordinates) in Safety-Maze, whereas 72D and 88D (LIDAR sensor outputs) in Safety-Goal and Safety-Push,
> * **Complexity of dynamics**: Safety-Maze has easy dynamics, yet the cars in Safety-Goal and Safety-Push need to maneuver around hazards and pillars,
> * **Types of reward functions**: Safety-Maze has a sparse reward function (1 for reaching the goal, and zero for all other transitions), whereas Safety-Goal and Safety-Push utilize the displacement-based reward functions in Safety-Gymnasium.
> * **Objects to interact with**: In Safety-Push, the car needs to learn that it needs to attach its front end to the box in the right angle and then push the box around pillars to the goal.
>
> We do not study Point Mass [1], because we already modified (Safety-Maze) another environment in [1] called Maze, which has similar dynamics (2D observation and action spaces) and objectives (pushing a point mass to a goal).
> Regarding Bipedal Walker Stump Tracks, we argue that this environment is not necessarily more challenging than Safety-Goal and Safety-Push, as the observation space is only 24-dimensional.
>
> **W2:** 'Trivial' method
>
> We strongly disagree with this comment. While SCG is based on a similar logic from CURROT, it has three novel features to achieve safer training and optimality at test time:
>
> 1. Cost constraint in the curriculum update in Eq. (4).
> 2. Safety-prioritization via generating source distributions over safe contexts in Eq. (5).
> 3. Annealing coefficient to tune prioritization ratio of safe or performant contexts in Eq. (5) and (6).
>
> Our ablation results in Figure 6 showcase that all these features result in a well-balanced, safe curriculum learning approach. Simply adding the cost constraint without safety-prioritization phase (SCG-NoPS and SCG-NoPPPS) may make the training safer than CURROT (see the bottom row of Figure 4a), but does not achieve optimality, as SCG-NoPS fails to yield zero cost in multiple runs and SCG-NoPPPS cannot achieve 100% success. Furthermore, not having annealing (SCG-NoAnn) causes higher CV regret and a lower success rate than SCG with all three features (see Figures 6a and 6c).
>
> **W3:** Many other curricular learning methods have different settings (?)
>
> It is unclear to us why the reviewer considers this as a weakness.
> What settings does the reviewer refer to?
> We would like to ask the reviewer to clarify their comment for us to respond to them.

---

> ### Author Response · Authors · 2024-11-20
> **Response to Reviewer Neqb - 2**
>
> **Q1:** A proper baseline should use a goal that cares about safety and performance. The score can be a weighted sum of both scores.
>
> One of the baselines we evaluate, NaiveSafeCURROT, already modifies CURROT by combining performance and safety into a single reward function to generate curricula, as explained in the paragraph before Section 6.1. Our experiments indicate that SCG reaches higher success rates than NaiveSafeCURROT in all environments, whereas NaiveSafeCURROT fails to yield optimal policies in multiple runs in all environments. At the same time, SCG yields low constraint violation. Therefore, we conclude that treating reward and cost separately results in better performance and safer training.
>
> Furthermore, it is already commonly accepted in the safe RL literature that simply combining the costs and rewards into a single reward function does not lead to safe behavior in many cases [2,3,4].
>
> **Q2:** Have you thought about the possibility of making more mistakes first and learning those mistakes can avoid future mistakes
>
> The ablation study in Section 6.3 responds to this question by evaluating SCG-NoPS, SCG without a safety prioritization phase. SCG-NoPS skips Phase 1 and directly looks for performant contexts at the expense of collecting cost, violating the cost constraint. Figure 6 indicates that SCG-NoPS not only yields higher CV regret but also fails to yield safe policies (zero cost in target contexts) in multiple runs.
>
> - **References**
>     1. Klink, P., Yang, H., D’Eramo, C., Peters, J., & Pajarinen, J. (2022, June). Curriculum reinforcement learning via constrained optimal transport. In International Conference on Machine Learning (pp. 11341-11358). PMLR.
>     2. Ray, A., Achiam, J., and Amodei, D. (2019). Benchmarking safe exploration in deep reinforcement learning.
>     3. Roy, J., Girgis, R., Romoff, J., Bacon, P., and Pal, C. J. (2022). Direct behavior specification via constrained reinforcement learning. *ICML*, 18828–18843.
>     4. Kamran, D., Simão, T. D., Yang, Q., Ponnambalam, C. T., Fischer, J., Spaan, M. T. J., and Lauer, M. (2022). A modern perspective on safe automated driving for different traffic dynamics using constrained reinforcement learning. *ITSC*, 4017–4023.

---

> ### Author Response · Authors · 2024-11-25
> **A kind reminder**
>
> In our rebuttal, we have addressed the reviewer's concerns over the contributions of our work and the environments experimented with. Our global response provides an additional ablation study to strengthen our claims and describes minor clarifications/changes in our submission to address the reviewer's concerns and improve the presentation.
>
> We hope our rebuttal and the new version of our submission have sufficiently addressed the reviewer's comments and questions. We are happy to address any further comments/questions the reviewer has during the discussion phase. If not, we kindly request the reviewer to consider revising their score, as one of the reviewers changed their score towards acceptance upon reading our rebuttal.

---

### Author Response · Authors · 2024-11-20
**Global Response**

We appreciate all reviewers for their thought-provoking comments and helpful questions. Thanks to them, we added new ablation studies and visualizations to support our claims better and made the following subtle yet critical changes:

* We ran [an ablation study on SCG parameters](https://drive.google.com/file/d/1ecIP19DZQIuFzXOWAGLn8PmdP4uNRpYU/view?usp=sharing) (see Appendix F.3)
    * Target Gaussian mixture model weight ratio $\alpha$, i.e., the target for annealing $\alpha_k$, determines the smooth transition between safety and performance prioritization phases.
    * Number of annealing iterations $K_{ann}$, which specificies how many curriculum iterations the annealing of $\alpha_l$ to $\alpha$ takes.
    * Cost threshold $\tilde{D}$ in SCG update sets the maximum expected cost SCG allows a context under the new context distribution $\rho_k$.
In short, the results showcase that SCG yields lower constraint violation regret than CURROT in all experimented values of these parameters. However, the set of values in the paper yields zero cost and 100% success in target contexts.
* We added [a new visualization of curriculum progression](https://drive.google.com/file/d/1ecIP19DZQIuFzXOWAGLn8PmdP4uNRpYU/view?usp=sharing) plots to indicate goal tolerance via varying diameters.
* We clarified our problem statement by mentioning the misalignment problem (see Section 3.2).
* We clarified the statistics that box plots demonstrate (see the caption of Fig.4 in Section 6.1).
* We added a remark on algorithmic differences between SCG and CURROT (see the end of Section 5).
* We extended the discussion on environments of interest to include the motivation behind them (see Section 6).
* We extended the discussion on evaluated state-of-the-art curriculum learning approaches (see Appendix B).

The changes described above are colored in red in our new submission.

---

### Meta-Review · Area_Chair_nDRi · 2024-12-20

**Metareview:**

This paper proposes Safe Curriculum Generation (SCG), a method for incorporating safety constraints into curriculum learning for reinforcement learning. SCG aims to improve both performance and safety by prioritizing tasks with minimal safety violations in the early stages of training.

Strengths
-----------
- **Clear and well-written:** The paper is well-organized and presents the problem, related work, and proposed method in a clear and easy-to-understand manner.
- **Improved performance and safety:** SCG demonstrates superior performance compared to existing curriculum learning methods while also minimizing safety violations during training.
- **Thorough evaluation:** The paper provides a comprehensive evaluation of SCG on various benchmark environments, including Safety-Maze, Safety-Goal, and Safety-Push.

Weaknesses
---------------
- **Limited novelty:** The main weakness is the perceived lack of significant novelty. SCG is seen as a relatively straightforward extension of the existing CURROT algorithm by adding safety constraints.

- **Limited environment complexity:** The evaluation is limited to relatively simple environments, raising concerns about the scalability and applicability of SCG to more complex real-world scenarios.

- **Lack of comparison with other safe RL methods:** The paper primarily compares SCG with curriculum learning methods and their naive safe adaptations, but it lacks comparisons with other dedicated safe RL algorithms.

SCG presents a valuable contribution by addressing the safety concerns in curriculum learning. Initially, reviewers mostly agreed that the paper needs further development to strengthen its novelty, demonstrate scalability to more complex environments, and provide comparisons with a wider range of safe RL methods. However, after discussion, most reviewers' concerns have been resolved. The improved revision and provided comments made this paper sufficiently good for publication.

**Additional Comments On Reviewer Discussion:**

Initially, reviewers mostly agreed that the paper had several limitations about the empirical analysis and the novelty. The discussion has been crucial to address these points and convince most of them to raise their score. Only one reviewer kept their score, but did not acknowledge the rebuttal nor engaged into discussion. Because of this, I have downweighted their review.

---

### Decision · Program_Chairs · 2025-01-22

Accept (Poster)